# NEMESIS: NORMALIZING THE SOFT-PROMPT VECTORS OF VISION-LANGUAGE MODELS

**Shuai Fu**[1], **Xiequn Wang**[1], **Qiushi Huang**[1,2], **Yu Zhang**[1,*]

[1]Department of Computer Science and Engineering, Southern University of Science and Technology
[2]Computer Science Research Centre, University of Surrey
{fus.jayce, wangxiequn, yu.zhang.ust}@gmail.com
qiushi.huang@surrey.ac.uk

## ABSTRACT

With the prevalence of large-scale pretrained vision-language models (VLMs), such as CLIP, soft-prompt tuning has become a popular method for adapting these models to various downstream tasks. However, few works delve into the inherent properties of learnable soft-prompt vectors, specifically the impact of their norms to the performance of VLMs. This motivates us to pose an unexplored research question: "Do we need to normalize the soft prompts in VLMs?" To fill this research gap, we first uncover a phenomenon, called the **Low-Norm Effect** by performing extensive corruption experiments, suggesting that reducing the norms of certain learned prompts occasionally enhances the performance of VLMs, while increasing them often degrades it. To harness this effect, we propose a novel method named **N**ormalizing th**e** soft-pro**m**pt **v**ectors of vi**si**on-language model**s** (**Nemesis**) to normalize soft-prompt vectors in VLMs. To the best of our knowledge, our work is the first to systematically investigate the role of norms of soft-prompt vector in VLMs, offering valuable insights for future research in soft-prompt tuning. The code is available at https://github.com/ShyFoo/Nemesis.

## 1 INTRODUCTION

In the age of large-scale pretrained vision-language models (VLMs), such as CLIP (Radford et al., 2021), Flamingo (Alayrac et al., 2022), and BLIP (Li et al., 2022), soft-prompt-based methods, also known as prompt-tuning, have emerged as a dominant approach for adapting these models to a wide range of downstream tasks. For instance, Zhou et al. (2022b) propose a Context Optimization (CoOp) method to learn soft prompts in a continuous space of CLIP for image classification tasks. Additionally, Rao et al. (2022) and Du et al. (2022) also employ prompt-tuning to address dense prediction and open-vocabulary object detection tasks, respectively.

Recent research in the field of VLMs has been primarily focused on enhancing model performance through the alignment of visual and textual features. For instance, in (Lu et al., 2022), the weight distribution of output embeddings is estimated, while Zang et al. (2022) propose a joint optimization approach for prompts across multiple modalities. Additionally, Chen et al. (2023) employs optimal transport techniques. To interpret learned soft-prompt vectors, Zhou et al. (2022b) and Chen et al. (2023) map them to the nearest words within the embedding space. More recently, Oymak et al. (2023) delves into the role of attention mechanisms in prompt-tuning, specifically within the context of a one-layer attention network.

While considerable advancements have been made in soft-prompt-based techniques for VLMs, scant attention has been paid to their intrinsic properties, specifically the norms of learnable soft-prompt vectors. We argue that the norms of soft prompts are a crucial but overlooked attribute that significantly influences the performance of VLMs. This paper addresses an overlooked aspect and presents a research question "Do we need to normalize the soft prompts in VLMs?" To the best of our knowledge, there is no work to study this question.

---

*Corresponding author.

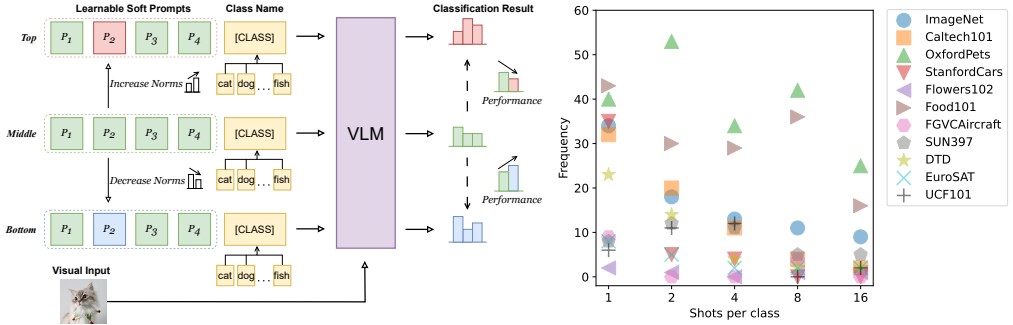

(a) *Top*: corrupted soft prompts with increased norms (b) The occurrence frequency of the Low-Norm leading to decreased performance; *Middle*: soft prompts Effect across 11 datasets. Each distinct color or learned by CoOp; *Bottom*: corrupted soft prompts with geometrical shape represents a different dataset. reduced norms resulting in enhanced performance.

Figure 1: A schematic diagram of the Low-Norm Effect. (a) The occurrence of the Low-Norm Effect in soft-prompt tuning VLMs. (b) The occurrence frequency of the Low-Norm Effect across 11 datasets commonly used in soft-prompt tuning VLMs.

To thoroughly investigate the role of soft-prompt vector norms in VLM performance, we introduce two corruption operations, *REPLACE* and *RESCALE*, to alter the norms in vectors learned by CoOp (Zhou et al., 2022b). Through the corruption of learned soft prompts, an intriguing phenomenon emerges: the reduction of norms at specific positions within these prompts enhances performance, whereas an increase in norms typically results in performance deterioration, as illustrated in Figure 1(a). We term this previously uncovered phenomenon the **Low-Norm Effect**.

Figure 1(b) explores the prevalence of the Low-Norm Effect across 11 widely-used prompt-tuning VLM datasets. Notably, the Imagenet (Deng et al., 2009), OxfordPets (Parkhi et al., 2012), and Food101 (Bossard et al., 2014) datasets show a higher frequency of this effect compared to datasets like Flowers102 (Nilsback & Zisserman, 2008). Moreover, we observe a negative correlation between the occurrence frequency of the Low-Norm Effect and the number of shots, indicating that less training data tends to induce the effect. These discrepancies might be linked to potential degradation in soft-prompt methods with limited data resources, thereby affecting model performance. Addressing the Low-Norm Effect remains challenging due to its inconsistent manifestation across datasets.

To exploit the Low-Norm Effect for enhancing the performance of VLMs, we propose a method called **N**ormalizing th**e** soft-pro**m**pt v**e**ctors of vi**si**on-language model**s** (**Nemesis**). We employ a Position-Uniform Normalization (PUN) loss to regulate the norms of all prompt vectors. This approach can be easily integrated into existing soft-prompt methods with negligible computation costs. However, the PUN loss may degrade the performance since it may normalize soft-prompt vectors that are unaffected by the Low-Norm Effect.

To handle this, the Position-Aware Normalization (PAN) loss is proposed as a refined substitute for the PUN loss. Specifically, a pre-inference step is introduced before each training batch to identify positions that are likely to induce the Low-Norm Effect. The pre-inference step applies corruption operations to generate multiple sets of corrupted prompts at distinct positions and then evaluates these corrupted prompts against their non-corrupted counterparts. This allows for the identification of positions that induce the Low-Norm Effect, followed by selective normalization of those positions, ultimately improving the performance. Extensive experiments demonstrate that the proposed Nemesis method could help boost the performance of soft-prompt-based VLMs in various tasks.

## 2 LOW-NORM EFFECT

In this section, we examine how the norms of learned prompt vectors influence the performance of VLMs and identify the Low-Norm Effect. To achieve that, we conduct extensive corruption

experiments by altering the norms of learned prompt vectors and applying two corruption operations (i.e., *REPLACE* and *RESCALE*) proposed in Section 3.2.

Specifically, following (Zhou et al., 2022b), we train the CoOp model and obtain the learned prompt vectors with a length of $L$. Then we corrupt these soft-prompt vectors at a single position each time and record the change of the performance and norms of soft prompts. To reduce the effect of experimental randomness, we average outcomes over five distinct runs using different seeds. Implementation details and results can be found in Appendix A.2.3.

The corruption experiments reveal a interesting phenomenon: reducing the norms of soft prompts at specific positions enhances the performance, while increasing them could degrade the performance, which is termed Low-Norm Effect. This phenomenon responds to the research question posed in Section 1 by uncovering the previously unexplored Low-Norm Effect in prompt-tuning VLMs. It also motivates the proposed Nemesis method for soft prompt normalization. Moreover, Section 4.6 offers a plausible explanation for the occurrence of the Low-Norm Effect and provides insights into the effectiveness of the proposed method. We believe our discovery can offer valuable insights for future research in soft-prompt tuning, laying the groundwork for potential advancements.

## 3 METHODOLOGY

In this section, we introduce the proposed **Nemesis** method. We begin with a review of the CoOp method (Zhou et al., 2022b) and subsequently introduce two key corruption operations, *REPLACE* and *RESCALE*. Finally, we present the entire method.

### 3.1 A REVISIT OF PROMPT-TUNING VISION-LANGUAGE MODELS

Over the years, pretrained VLMs have demonstrated impressive generalization performance in zero-shot open-world visual recognition, wherein the model can perform a task without undergoing explicit training. One typical paradigm is CLIP (Radford et al., 2021), which consists of an image encoder and a text encoder. CLIP is trained on approximately 400 million image-text pairs, contributing to its remarkable performance. Nevertheless, effectively fine-tuning these VLMs for downstream tasks remains a challenge, particularly when dealing with few-shot data, due to their massive parameters. The CoOp method addresses this issue by setting the templated context prompts (e.g. `This is a photo of {class-name}.`) as learnable vectors, which only requires fine-tuning these learnable vectors while keeping the pretrained VLMs frozen. For a downstream visual recognition task consisting of $C$ categories, the classification weights of one image can be defined by the similarity between the visual feature and the text features of all categories.

Formally, the image encoder and text encoder can be denoted by $f$ and $g$, respectively. Given an image $x$ along with its classification label $y$, the visual feature can be formulated as $\boldsymbol{f} = f(x)$, while the textual prompt of $i$-th class can be formulated as $\boldsymbol{t}_i = \{\boldsymbol{v}_1, \boldsymbol{v}_2, \boldsymbol{v}_j, ..., \boldsymbol{v}_L, \boldsymbol{c}_i\}$, where $\boldsymbol{v}_j$ and $\boldsymbol{c}_i$ denote the $j$-th soft-prompt vector and the word embedding of the class name, respectively. Then the $i$-th class textual feature can be denoted as $\boldsymbol{g}_i = g(\boldsymbol{t}_i)$. Given few-shot data, CoOp can learn the soft prompts $\mathbb{V}^{L \times D} = \{\boldsymbol{v}_1, \boldsymbol{v}_2, ..., \boldsymbol{v}_L\}$, where $L$ and $D$ denote the length of soft prompts and the dimension of prompt vectors, respectively, by minimizing the negative log-likelihood between the image feature $\boldsymbol{f}$ and its ground-truth textual feature $\boldsymbol{g}_y$ as

$$\mathcal{L}_{\text{CE}} = -\sum_{x \in \mathbf{X}} \log \frac{\exp(\text{sim}(\boldsymbol{f}, \boldsymbol{g}_y)/\lambda)}{\sum_{i=1}^{C} \exp(\text{sim}(\boldsymbol{f}, \boldsymbol{g}_i)/\lambda)}, \tag{1}$$

where $\lambda$ is a temperature parameter and $\text{sim}(\cdot, \cdot)$ denotes the cosine similarity function. After the training process, the text encoder $g$ encodes both the learned prompts $\mathbb{V}$ and the class embeddings to produce textual features for all classes.

### 3.2 CORRUPTION OPERATIONS

In this section, we introduce two corruption operations: *REPLACE* and *RESCALE*, which can be employed to corrupt the learned soft-prompt vectors.

For the *REPLACE* operation, we replace learned prompt vectors at a single position with a zero-mean Gaussian-distributed vector with fixed variance. Then, we can obtain a set of corrupted soft

prompts $\boldsymbol{t}_i^{re} = \{\boldsymbol{v}_1, \boldsymbol{v}_2, \boldsymbol{r}_j, ..., \boldsymbol{v}_L, \boldsymbol{c}_i\}$, where the $j$-th prompt vector is replaced with a random Gaussian vector $\boldsymbol{r}_j \sim \mathcal{N}(\mu\mathbf{1}, \sigma^2\mathbf{I})$, where $\mathbf{1}$ denotes a vector of all ones with an appropriate size, $\mathbf{I}$ denotes an identity matrix with an appropriate size, $\mu$ denotes the mean for each dimension, and $\sigma$ denotes the standard deviation for each dimension.

For the *RESCALE* operation, we rescale the value of $j$-th prompt vector using various rescaling factors. Then, we can obtain a set of corrupted soft prompts $\boldsymbol{t}_i^{sc} = \{\boldsymbol{v}_1, \boldsymbol{v}_2, s \times \boldsymbol{v}_j, ..., \boldsymbol{v}_L, \boldsymbol{c}_i\}$, where $s$ is a rescaling factor.

In essence, both corruption operations can be regarded as strategies to modify the soft prompts, subsequently altering their norms. By changing soft prompts, the textual features generated by $g$ are influenced, consequently impacting final predictions. Specifically, given corrupted soft prompts, the prediction probability can be calculated as

$$p_{csp}(y|x) = \frac{\exp(\text{sim}(\boldsymbol{f}, \boldsymbol{g}_y^{csp})/\lambda)}{\sum_{i=1}^{C} \exp(\text{sim}(\boldsymbol{f}, \boldsymbol{g}_i^{csp})/\lambda)}, \tag{2}$$

where $\boldsymbol{g}^{csp}$ can be either $g(\boldsymbol{t}_i^{re})$ or $g(\boldsymbol{t}_i^{sc})$. Note that the two corruption operations we proposed can be applied to any learned soft prompts, thereby facilitating research in the field of investigating the impact of soft-prompt norms to the model performance.

## 3.3 THE NEMESIS METHOD

To handle the Low-Norm Effect during prompt-tuning VLMs, we propose two losses for normalizing the norms of soft prompts: Position-Uniform Normalization (PUN) loss and Position-Aware Normalization (PAN) loss. In the experiments, they are separated as an individual regularization item, which is added to the standard soft-prompt tuning process.

Generally, given a set of soft prompts $\mathbb{V}^{L \times D} = \{\boldsymbol{v}_1, \boldsymbol{v}_2, ..., \boldsymbol{v}_L\}$, we can calculate their norms as $\frac{1}{M} \sum_{j=1}^{L} \alpha_j \|\boldsymbol{v}_j\|_p$, where $M$ denotes the number of non-zero values in the set $\{\alpha_1, \alpha_2, \ldots, \alpha_L\}$ and $\|\cdot\|_p$ denotes the $\ell_p$-norm of a vector. Unless otherwise specified, we use the $\ell_2$ norm by default.

For the PUN loss, all elements of the set $\{\alpha_1, \alpha_2, \ldots, \alpha_L\}$ are set to the same value, imposing an equal weight on the norms of soft prompts at all positions. Hence, this loss can be formulated as

$$\mathcal{L}_{\text{PUN}} = \frac{1}{M} \sum_{j=1}^{L} \alpha_j \|\boldsymbol{v}_j\|_p, \tag{3}$$

where $\alpha_j = \omega$ for $j = 1, \ldots, L$. Here $\omega$ is a scaling coefficient that controls the normalization strength. However, normalizing prompt vectors at positions unaffected by the Low-Norm Effect may not yield performance improvement. This is because it could potentially restrict the weight updates of soft prompts at these positions. Hence, it is necessary to tackle the Low-Norm Effect at each prompting position and dynamically adjust $\alpha_j$ during training.

On the other hand, if the Low-Norm Effect can be explicitly recognized during the training process, we can effectively address this issue and enhance the efficacy of soft-prompt learning. To achieve this, we incorporate an additional inference process prior to each batch training iteration to identify the prompting positions that induce the Low-Norm Effect.

Similar to corruption experiments, we initially set a rescaling factor, denoted by $\tau$, to induce the Low-Norm Effect, where $\tau$ is a positive real number less than 1. Then we apply the *RESCALE* operation on a normal soft prompt $\mathbb{V}$ to generate $N$ sets of corrupted prompts at distinct prompting positions $\{\mathbb{V}_{l_1}, \mathbb{V}_{l_2}, \ldots, \mathbb{V}_{l_n}, \ldots, \mathbb{V}_{l_N}\}$, where $l_n$ denote corrupted positions. Note that for each training batch, we randomly select $N$ distinct positions from the set of $L$ positions $\mathscr{L} = \{1, 2, \ldots, L\}$. Formally, the conditions $1 \leq l_1 \neq l_2 \ldots \neq l_n \ldots \neq l_N \leq L$ and $N \leq L$ ensure that the positions of rescaled prompt vectors for each set of corrupted prompts are distinct from each other.

By having a set of images $\mathscr{X}$ with a batch size of $B$ as well as their ground-truth labels $\mathscr{Y} = (y_1, \ldots, y_B)$, and a hybrid prompt set $\mathscr{V}^{(N+1) \times L \times D} = \{\mathbb{V}, \mathbb{V}_{l_1}, \mathbb{V}_{l_2}, \ldots, \mathbb{V}_{l_N}\}$, where $\mathbb{V}$ is the original prompt and others are corrupted prompts, we can obtain a set of label predictions $\hat{\mathscr{Y}}^{(N+1) \times B}$

of VLMs, where the first row corresponds to the batch predictions for the original prompt and the next $N$ rows represent the batch predictions for its corrupted counterparts.

By calculating prediction scores for all prompts and comparing them, we can identify corrupted prompts at specific positions that yield more accurate predictions than their original counterparts. This observation indicates the occurrence of the Low-Norm Effect during training. As a result, only those soft prompts at the prompting positions that induce the Low-Norm Effect will be normalized during this training batch. Hence, the PAN loss can be defined as

$$\mathcal{L}_{\text{PAN}} = \frac{1}{M} \sum_{k=l_1}^{l_N} \alpha_k \|\boldsymbol{v}_k\|_p,$$ (4)

where $\alpha_k$ equals $\omega$ if $\sum_{b=1}^{B} \mathbb{1}(\hat{y}_{b,k} = y_b) > \sum_{b=1}^{B} \mathbb{1}(\hat{y}_b = y_b)$ and otherwise 0 for $k = l_1, l_2, \ldots, l_N$. $\hat{y}_{b,k}$ refers to the prediction for the $b$-th sample in the batch, using the soft prompts which have been corrupted at $k$-th position, while $\hat{y}_b$ represents the prediction for the $b$-th sample in the batch, using the original, uncorrupted soft prompts. The detailed algorithm by adopting $\mathcal{L}_{\text{PAN}}$ can be found in Section A.1.

To sum up, for a training batch, we can optimize soft prompts by minimizing the following total objective as

$$\mathcal{L} = \mathcal{L}_{\text{CE}} + \beta \mathcal{L}_{\text{PUN}} + (1 - \beta) \mathcal{L}_{\text{PAN}},$$ (5)

where $\beta$, which equals either 0 or 1, corresponds to two variants of the proposed Nemesis method.

## 4 EXPERIMENTS

In this section, extensive experiments are conducted to evaluate the proposed **Nemesis** method, including comparison with CoOp (Zhou et al., 2022b) on few-shot image classification tasks and domain generalization tasks, comparison with CoCoOp (Zhou et al., 2022a) in the base-to-new generalization setting. Additionally, we conduct an in-depth impact analysis on VLM performance due to the norms of soft prompts, explore the method's extensibility to other soft-prompt tuning approaches, and assess the computational efficiency.

### 4.1 DATASETS

For few-shot image classification experiments and base-to-new generalization tasks, we follow the experimental setting of CoOp and CoCoOp, respectively, and conduct experiments on 11 visual classification datasets, including Caltech101 (Fei-Fei et al., 2004) and ImageNet (Deng et al., 2009) for object recognition, EuroSAT (Helber et al., 2019) for satellite image recognition, DTD (Cimpoi et al., 2014) for texture recognition, UCF101 (Soomro et al., 2012) for action recognition, SUN397 (Xiao et al., 2010) for scene recognition, OxfordPets (Parkhi et al., 2012), FGVCAircraft (Maji et al., 2013), Food101 (Bossard et al., 2014), Flowers102 (Nilsback & Zisserman, 2008), and Stan-fordCars (Krause et al., 2013) for fine-grained recognition. Besides, ImageNet (Deng et al., 2009) and its variants, including ImageNet-A (Hendrycks et al., 2021b), ImageNet-R (Hendrycks et al., 2021a), ImageNetV2 (Recht et al., 2019), and ImageNet-Sketch (Wang et al., 2019), are used for the evaluation of domain generalization. Detailed descriptions of each dataset can be found in Appendix A.2.1.

### 4.2 IMPLEMENTATION DETAILS

For few-shot image classification experiments and domain generalization tasks, we compare our method with the baseline method CoOp, while CoCoOp is chosen as our baseline model in base-to-new generalization tasks. Following the few-shot evaluation protocol used in CoOp, we use a fixed number of training samples from each category (i.e. 1, 2, 4, 8, 16 shots per class). Besides, we follow the same training configurations as these baseline models, including training epochs, learning rate, and batch size, etc. All reported results are based on the average of five different seed runs. Bold denotes the best performance on each comparison setting. More implementation details and hyper-parameter settings can be found in Section A.2.2.

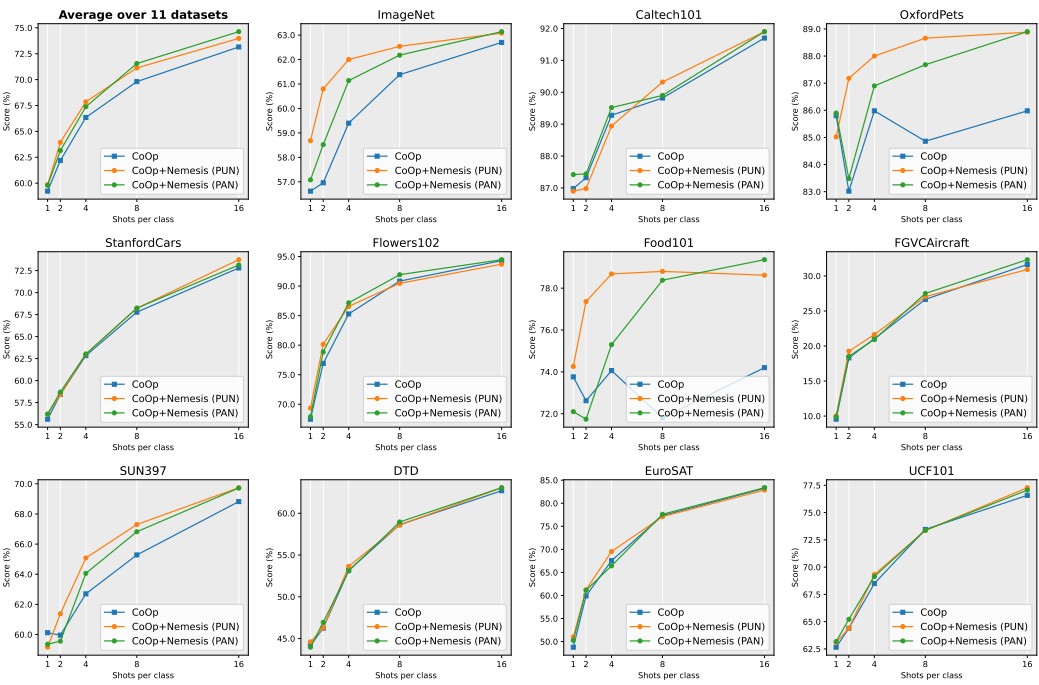

Figure 2: The few-shot recognition results of CoOp and CoOp+Nemesis (ours) on 11 datasets.

## 4.3 FEW-SHOT IMAGE RECOGNITION RESULTS

The experimental results of few-shot recognition are summarised in Figure 2. The blue, orange, and green lines represent CoOp, CoOp+Nemesis with the PUN loss, and CoOp+Nemesis with the PAN loss, respectively. In terms of average performance, both Nemesis methods outperform CoOp. Particularly, they achieved a large improvement over CoOp on the ImageNet, OxfordPets, Food101, and SUN397 datasets. This indicates that normalizing the soft prompts in VLMs can lead to better performance on these datasets that exhibit a more pronounced Low-Norm Effect. Taking the ImageNet dataset as an example, Nemesis with the PUN loss gains 2.06%, 3.84%, 2.6%, 1.16%, 0.38% performance boost over CoOp at 1, 2, 4, 8, 16 shots. Similarly, Nemesis with the PUN loss also shows performance improvements of 0.46%, 1.56%, 1.74%, 0.80%, and 0.44%. Moreover, it is evident that CoOp+Nemesis demonstrates enhanced robustness and superior performance on the Food101 and OxfordPets, compared with CoOp. Additionally, comparing Nemesis with the PUN loss, Nemesis with the PAN loss shows more robust performance at larger shot settings. All these performance comparisons demonstrate normalizing the soft prompts in VLMs can facilitate the effective learning of soft prompts for few-shot recognition. More detailed data and analysis of training process can be found in Appendix A.2.6.

## 4.4 EVALUATION OF GENERALIZATION PERFORMANCE

In this subsection, we conduct experiments to assess the generalization performance of the proposed method. All methods are trained on the ImageNet dataset with 16 shots per class and tested on four different ImageNet-based datasets. Table 1 reports the results of CoOp, CoOp+Nemesis (PUN), and CoOp+Nemesis (PAN). It is clear that CoOp+Nemesis outperforms CoOp consistently on both source and target domains, whether adopting the PUN loss or PAN loss, which suggests that Nemesis can improve CoOp's domain generalization abilities by normalizing the soft prompts in VLMs. Furthermore, we can observe that Nemesis using larger $\omega$ can achieve better transfer performance, implying that a stronger normalization of soft prompts could enhance the robustness of soft prompts to domain shifts. Comparing Nemesis using the PUN loss and Nemesis using the PAN loss, despite that the latter achieves better performance on the source domain, its performance on target domains is inferior to the former. We argue that this may arise due to the PAN loss excessively prioritizing to identify and address the Low-Norm Effect within intra-domain data, which could compromise its generalization capability. The results of base-to-new experiments can be found in Appendix A.2.4.

Table 1: Comparison of CoOp and CoOp+Nemesis (ours) in the domain generalization setting.

| Method | Source | Target | | | |
|---|---|---|---|---|---|
| | ImageNet | -V2 | -Sketch | -A | -R |
| CoOp | 62.70 | 55.04 | 32.64 | 22.44 | 54.60 |
| CoOp+Nemesis (PUN, $\omega = 1$) | 62.96 | 55.48 | 33.64 | 22.96 | 55.72 |
| CoOp+Nemesis (PUN, $\omega = 10$) | 63.08 | **55.50** | **34.34** | **23.40** | **56.78** |
| CoOp+Nemesis (PAN, $\omega = 1$) | **63.28** | 55.16 | 32.70 | 22.48 | 54.72 |
| CoOp+Nemesis (PAN, $\omega = 10$) | 63.14 | 55.20 | 33.50 | 23.30 | 56.68 |

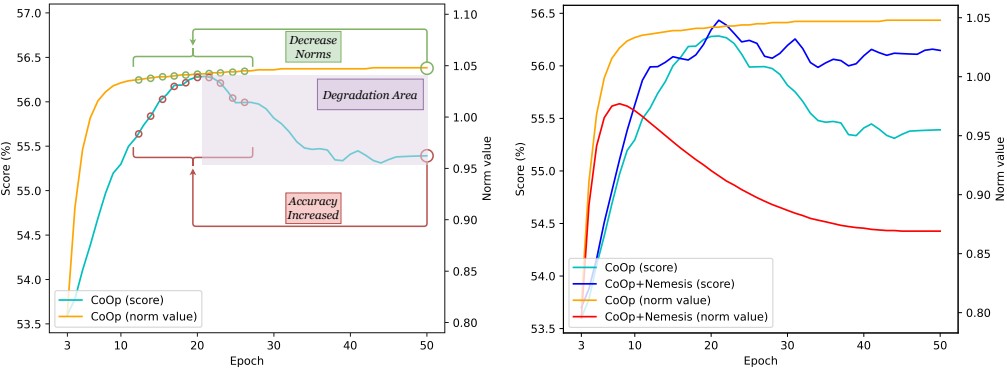

(a) The occurrence of the Low-Norm Effect in CoOp. (b) Comparison of CoOp and CoOp+Nemesis (ours).

Figure 3: Analysis of the impact of soft prompt norms on the model's performance using the StanfordCars dataset as an example. The X-axis, Y1-axis, and Y2-axis represent training epochs, test accuracy, and average norm value of all soft-prompt vectors, respectively.

It also can be found that the generalization performance from base classes to unseen classes can be improved by normalizing the soft prompts in VLMs.

## 4.5 IN-DEPTH STUDIES ON THE LOW-NORM EFFECT IN VLMS

In this section, we aim to provide plausible explanations for the occurrence of the Low-Norm Effect and the effectiveness of the proposed method Nemesis.

From Figure 3(a), it is apparent that the norms of soft prompts in CoOp first increase and then level off, while test accuracy falls into degradation as norms slowly flatten out. By performing corruption operations that decrease the norms of prompt vectors, the last green circle may be pushed away from the degradation area and get closer to those small green circles that demonstrate superior performance. This could be regarded as a plausible explanation for the occurrence of the Low-Norm Effect: those corrupted soft prompts that demonstrate superior performance than their original counterparts may be precisely one of those small circles. Moreover, this figure may unveil a potential correlation between the time when prompt learning starts to degrade and the time when the norm of soft prompts begins to stabilize. We leave this to future research.

From Figure 3(b), different from the observed norm variation pattern in CoOp, CoOp+Nemesis (ours) exhibits a distinct trend where norms initially increase, followed by a subsequent decrease, and eventually stabilize. Furthermore, the test accuracy exhibits a consistent upward trend before reaching a plateau, whereas a declining trend is observed in CoOp. This implies that our method can delay the time point where soft prompts tend to plateau during the learning process, thereby reducing the probability of learning degradation.

## 4.6 EXTENDIBILITY ANALYSIS

To analyze the extensibility of the proposed approach Nemesis, we apply the proposed method Nemesis to other soft prompt-tuning methods on few-shot recognition experiments. PLOT (Chen

Table 2: The few-shot recognition results of PLOT and PLOT+Nemesis (ours) on several datasets.

| Dataset | Method | 1 shot | 2 shots | 4 shots | 8 shots | 16 shots |
|---------|--------|--------|---------|---------|---------|----------|
| Caltech101 | PLOT | 88.80 | 87.94 | 89.44 | 90.46 | 92.16 |
| | PLOT+Nemesis (PUN) | 89.14 | **88.02** | 89.74 | 90.38 | 92.10 |
| | PLOT+Nemesis (PAN) | **89.16** | 87.98 | **89.86** | **90.80** | **92.30** |
| DTD | PLOT | 46.42 | 51.52 | 56.10 | 60.66 | 64.16 |
| | PLOT+Nemesis (PUN) | 46.64 | **51.58** | **56.22** | 60.94 | 64.46 |
| | PLOT+Nemesis (PAN) | **46.65** | 51.40 | 56.08 | **61.56** | **64.68** |
| Flowers102 | PLOT | 71.06 | 81.38 | 87.68 | 92.58 | 94.92 |
| | PLOT+Nemesis (PUN) | 70.94 | 81.44 | **88.42** | 92.86 | 95.32 |
| | PLOT+Nemesis (PAN) | **71.40** | **81.96** | 88.24 | **93.06** | **95.46** |
| Average | PLOT | 68.76 | 73.61 | 77.74 | 81.23 | 83.75 |
| | PLOT+Nemesis (PUN) | 68.91 | 73.68 | **78.13** | 81.39 | 83.96 |
| | PLOT+Nemesis (PAN) | **69.07** | **73.78** | 78.05 | **81.81** | **84.15** |

Table 3: Ablation studies about the PAN loss on the FGVCAircraft dataset. $N$ denotes the number of corruption positions, while $w/o\ selection$ means the prompting positions where the soft prompts are directly normalized, without the selection process within the PAN loss.

| Method | 1 shot | 2 shots | 4 shots | 8 shots | 16 shots |
|--------|--------|---------|---------|---------|----------|
| CoOp | 9.56 | 18.30 | 21.04 | 26.66 | 31.64 |
| CoOp+Nemesis ($N = 1, w/o\ selection$) | 9.60 | 19.34 | 21.64 | 26.98 | 30.92 |
| CoOp+Nemesis ($N = 1$)* | 9.88 | 18.52 | 20.94 | 27.50 | 32.32 |
| CoOp+Nemesis ($N = 2, w/o\ selection$) | 9.50 | 18.72 | 21.72 | 27.22 | 30.90 |
| CoOp+Nemesis ($N = 2$) | 9.74 | 18.38 | 21.18 | 27.36 | **32.46** |
| CoOp+Nemesis ($N = 4, w/o\ selection$) | 10.04 | 19.20 | 22.10 | 27.36 | 31.26 |
| CoOp+Nemesis ($N = 4$) | 9.44 | 19.02 | 21.26 | 27.52 | 32.10 |
| CoOp+Nemesis ($N = 8, w/o\ selection$) | 9.62 | 19.28 | 21.58 | 26.92 | 30.88 |
| CoOp+Nemesis ($N = 8$) | 9.30 | 19.32 | 21.66 | 27.36 | 32.02 |
| CoOp+Nemesis ($N = 16, w/o\ selection$) | 9.82 | 19.24 | 21.52 | 26.98 | 31.12 |
| CoOp+Nemesis ($N = 16$) | **10.44** | **19.72** | **22.13** | **27.83** | 31.44 |

et al., 2023) is an ensemble-based prompt-tuning method and leverages optimal transport distance to achieve better alignment between multiple sets of soft prompts and image features. In this experiment, We set $\omega$ as 0.1 by default. Table 2 provides partial results for comparing PLOT and PLOT+Nemesis (our approach). In comparison to PLOT, PLOT+Nemesis demonstrates enhanced recognition performance for all shots. This verifies that ensemble-based prompt-tuning methods can also benefit from normalizing the soft prompts in VLMs. Furthermore, we discuss other applicable scenarios of the proposed method. More details can be found in Appendix A.2.8.

## 4.7 ABLATION STUDIES AND HYPER-PARAMETER ANALYSIS

The results of ablation studies about the PAN loss are shown in Table 3, where $w/o\ selection$ is analogous to utilizing the PUN loss to normalize soft prompts at $N$ random positions and * means the default setting in few-shot recognition tasks. Overall, increasing $N$ typically leads to improved performance, except for the case of 16 shots, where performance initially increases and then decreases with $N$. Besides, comparing the PAN loss without considering the selection process, the proposed selection of prompting positions that are used for normalization based on performance variation demonstrates a more robust performance.

Furthermore, Table 4 provides the results of combining the two proposed losses. We can observe that $\beta$=0.3 and $\beta$=0.1 achieve the best results under 1-shot and 16-shots, respectively. This suggests that the PAN loss plays a dominant role, while the PUN loss provides assistance, which can lead to improved performance. Besides, the discussion about the computation costs raised by the two proposed losses can be found in Appendix A.2.7.

Moreover, the hyper-parameter analysis about $\omega$, $\tau$, and norm types used in the normalization losses can be found in Tables A5, A6, and A7, respectively. The outcome data demonstrate that the pro-

Table 4: Ablation studies about $\beta$.

| | $\beta$=0 | $\beta$=0.1 | $\beta$=0.3 | $\beta$=0.5 | $\beta$=0.7 | $\beta$=0.9 | $\beta$=1 |
|---|---|---|---|---|---|---|---|
| 1-shot | 59.33 | 59.45 | **59.54** | 59.29 | 59.26 | 59.37 | 59.42 |
| 16-shots | 74.20 | **74.39** | 73.37 | 74.02 | 73.93 | 73.79 | 73.47 |

posed method Nemesis, is resilient when different norm types and values of $\tau$ are used. Additionally, we observe that a larger value of $\omega$ generally yields better performance for small shots, whereas a smaller value of $\omega$ performs well for large shots. This finding combined with the results of corruption experiments presented in Section 2, implies that soft prompts with a higher occurrence frequency necessitate stronger normalization. Therefore, determining the appropriate size of $\omega$ is also one of the potential research directions for the future.

## 5 RELATED WORK

### 5.1 VISION-LANGUAGE MODELS PRE-TRAINING

Vision-language models (VLMs), usually consisting of a visual module and a language module, are expected to explore the semantic connections between images and texts. With large-scale image-text pairs which are available on the internet, an increasing number of pre-trained VLMs (Radford et al., 2021; Cui et al., 2022; Yao et al., 2021) have been proposed. These pre-trained VLMs can capture deep vision-language semantic correspondence by using various vision-language objectives, including contrastive objective (Radford et al., 2021; Cui et al., 2022; Singh et al., 2022; Yao et al., 2021; Zhong et al., 2022), generative objective (Cui et al., 2022; Singh et al., 2022), and alignment objective (Yao et al., 2021; Zhong et al., 2022). Moreover, these pre-trained VLMs also show strong capacities for generalization and can perform zero-shot predictions (without fine-tuning models) on a wide range of downstream tasks, such as image classification (Radford et al., 2021), visual question answering (Alayrac et al., 2022), and text-guided image generation (Avrahami et al., 2022).

### 5.2 PARAMETER-EFFICIENT FINE-TUNING

Parameter-efficient fine-tuning (PEFT) methods serve as a crucial approach for adapting pretrained models, particularly within the Natural Language Processing (NLP) domain. Among these, prompt-tuning (Lester et al., 2021; Jiang et al., 2023) has gained attention for optimizing task-specific prompt embeddings, providing performance similar to full parametric fine-tuning but with fewer tunable parameters. Similarly, prefix-tuning (Li & Liang, 2021) extends this concept by optimizing a sequence of prefixes at each transformer layer, thereby augmenting the set of tunable parameters marginally, while P-tuning (Liu et al., 2022) incorporates manually designed patterns to intersperse learned prompts within the input embeddings. Inspired by these PEFT methods of NLP, this technique has been successfully extended to VLMs. For instance, CoOp (Zhou et al., 2022b) and its variants (Zhou et al., 2022a) apply CLIP (Radford et al., 2021) to few-shot visual recognition tasks by replacing hard-crafted prompts with learnable soft-prompt vectors. In addition, adapter-tuning (Gao et al., 2023), which allows for fine-tuning a part of the network or fine-tuning an extra network, are another research direction of PEFT method of VLMs. Distinctively, the proposed method, Nemesis, first provides empirical evidence that normalizing soft-prompt vectors of VLMs can help improve performance.

## 6 CONCLUSION

In this paper, we are the first to examine the impact of soft prompts' norms on the performance of VLMs. We conduct extensive corruption experiments using two specially designed operations and discover the Low-Norm Effect. To harness this phenomenon, we introduce Nemesis, a method for normalizing soft prompts during soft-prompt tuning. In general, Nemesis can be incorporated into any soft-prompt-based methods, even other PEFT methods, such as prefix-tuning, and P-tuning. We hope our findings and proposed method can provide new insights and facilitate future research on these fields.

ACKNOWLEDGMENTS

This work is supported by NSFC key grant under grant no. 62136005, NSFC general grant under grant no. 62076118, and Shenzhen fundamental research program JCYJ20210324105000003.

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

# A  APPENDIX

## A.1  METHOD DETAILS

During the training process, the proposed method Nemesis adopting the Position-Aware Normalization (PAN) loss would involve an additional inference process to determine those prompting positions inducing the Low-Norm Effect. On the other side, Nemesis does not change the inference process of original soft-prompt-based methods. Hence, here we only provide more details of the training process for Nemesis adopting the PAN loss in this subsection. Its iterative optimization procedure is summarized in Algorithm 1.

---

**Algorithm 1:** The training process of Nemesis adopting the PAN loss.

**Input:** The training data $\{x, y\} \in \{\mathbf{X}, \mathbf{Y}\}$, pretrained CLIP model $f$ and $g$, the threshold $\tau$ inducing the Low-Norm Effect, the number of corrupted positions $N$ for soft prompts, the scaling coefficient $\omega$ for the norm of soft-prompt vectors, the total training epochs $E$, and the training batch size $B$.

**Output:** The parameters of soft prompts $\mathbb{V}^{L \times D} = \{\boldsymbol{v}_1, \boldsymbol{v}_2, ..., \boldsymbol{v}_L\}$.

1   **Function** update_alpha($\mathbb{V}$, $\mathscr{X}$, $\mathscr{Y}$, $\tau$, $\omega$, $N$):

2      Randomly select $N$ distinct positions $\{l_1, l_2, \ldots, l_N\}$ from the set of $L$ positions $\{1, 2, \ldots, L\}$ to generate the corrupted prompt set;

3      Obtain the corrupted prompt set $\mathscr{V}^{(N+1) \times L \times D} = \{\mathbb{V}, \mathbb{V}_{l_1}, \mathbb{V}_{l_2}, \ldots, \mathbb{V}_{l_N}\}$, where $\mathbb{V}$ is the original prompt;
     `// Enabling the process of inference`

4      Generate a visual feature set $\mathscr{F}^{B \times C}$ of with the visual encoder $f(\mathscr{X})$;

5      Generate a corrupted textual feature set $\mathscr{G}_i^{B(N+1) \times C}$ of each class with the textual encoder $g(\text{concat}\{\mathscr{V}, \boldsymbol{c}_i\})$, where $\boldsymbol{c}_i$ denotes the word embedding of $i$-th class name;

6      Calculate the similarity between visual and textual feature $S_i = \mathscr{G}_i \mathscr{F}^\top$ of each class;
     `// Terminating the process of inference`

7      Obtain the classification predictions $\hat{\mathscr{Y}} = \begin{pmatrix} \hat{y}_1 & \hat{y}_2 & \cdots & \hat{y}_B \\ \hat{y}_{1,l_1} & \hat{y}_{2,l_1} & \cdots & \hat{y}_{B,l_1} \\ \vdots & \vdots & \ddots & \vdots \\ \hat{y}_{1,l_N} & \hat{y}_{2,l_N} & \cdots & \hat{y}_{B,l_N} \end{pmatrix}$;

8      Calculate a set of prediction performance:
     $\left( \sum_{b=1}^{B} \mathbb{1}(\hat{y}_b = y_b) \quad \sum_{b=1}^{B} \mathbb{1}(\hat{y}_{b,l_1} = y_b) \quad \cdots \quad \sum_{b=1}^{B} \mathbb{1}(\hat{y}_{b,l_N} = y_b) \right)$;

9      **for** $l_n$ to $\{l_1, l_2, \ldots, l_N\}$ **do**

10          **if** $\sum_{b=1}^{B} \mathbb{1}(\hat{y}_{b,l_n} = y_b) > \sum_{b=1}^{B} \mathbb{1}(\hat{y}_b = y_b)$ **then**

11             $\alpha_{l_n} = \omega$

12          **else**

13             $\alpha_{l_n} = 0$

14      **return** $\{\alpha_{l_1}, \alpha_{l_2}, \ldots, \alpha_{l_N}\}$

15 Initialize $\mathbb{V}$

16 **for** $e$ to $E$ **do**

17      **for** each training mini-batch $\mathscr{X}$, $\mathscr{Y}$ **do**

18          Obtain $\{\alpha_{l_1}, \alpha_{l_2}, \ldots, \alpha_{l_N}\}$ using update_alpha($\mathbb{V}$, $\mathscr{X}$, $\mathscr{Y}$, $\tau$, $\omega$, $N$);

19          Calculate the PAN loss $\mathcal{L}_{\text{PAN}} = \frac{1}{M} \sum_{j=l_1}^{l_N} \alpha_j \cdot \|\boldsymbol{v}_j\|_p$;

20          Generate the visual feature set $\mathscr{F}$ and textual feature set $\mathscr{G}_i$ of each class;

21          Calculate the cross-entropy loss $\mathcal{L}_{\text{CE}} = - \sum_{\mathscr{X} \in \mathbf{X}} \log \frac{\exp(sim(\mathscr{F}, \mathscr{G}_y)/\lambda)}{\sum_{i=1}^{C} \exp(sim(\mathscr{F}, \mathscr{G}_i)/\lambda)}$;

22          Update $\mathbb{V}$ by minimizing $\mathcal{L}_{\text{CE}}$ and $\mathcal{L}_{\text{PAN}}$

---

Table A1: The detailed statistics of datasets.

| Dataset | Classes | Train | Test | Task |
|---------|---------|-------|------|------|
| Caltech101 (Fei-Fei et al., 2004) | 100 | 4,128 | 2,465 | Object recognition |
| ImageNet (Deng et al., 2009) | 1,000 | 1.28M | 50,000 | Object recognition |
| EuroSAT (Helber et al., 2019) | 10 | 13,500 | 8,100 | Satellite image recognition |
| DTD (Cimpoi et al., 2014) | 47 | 2,820 | 1,692 | Texture recognition |
| UCF101 (Soomro et al., 2012) | 101 | 7,639 | 3,783 | Action recognition |
| SUN397 (Xiao et al., 2010) | 397 | 15,880 | 19,850 | Scene recognition |
| OxfordPets (Parkhi et al., 2012) | 37 | 2,944 | 3,669 | Fine-grained pets recognition |
| FGVCAircraft (Maji et al., 2013) | 100 | 3,334 | 3,333 | Fine-grained aircraft recognition |
| Food101 (Bossard et al., 2014) | 101 | 50,500 | 30,300 | Fine-grained food recognition |
| Flowers102 (Nilsback & Zisserman, 2008) | 102 | 4,093 | 2,463 | Fine-grained flowers recognition |
| StanfordCars (Krause et al., 2013) | 196 | 6,509 | 8,041 | Fine-grained cars recognition |
| ImageNet-A (Hendrycks et al., 2021b) | 200 | N/A | 7,500 | Robustness of adversarial attack |
| ImageNet-R (Hendrycks et al., 2021a) | 200 | N/A | 30,000 | Robustness of multi-domains |
| ImageNetV2 (Recht et al., 2019) | 1,000 | N/A | 10,000 | Robustness of collocation |
| ImageNet-Sketch (Wang et al., 2019) | 1,000 | N/A | 50,889 | Robustness of sketch domain |

## A.2 EXPERIMENTAL DETAILS

### A.2.1 DATASETS

Following CoOp (Zhou et al., 2022b) and CoCoOp (Zhou et al., 2022a), the datasets we used include 11 datasets for few-shot visual recognition and base-to-new generalization tasks, as well as 4 ImageNet-based datasets for the evaluation of domain generalization. The details of each dataset can be found in Table A1, including dataset name, the number of classes, the number of training and testing samples, as well as the type of visual task.

### A.2.2 HYPER-PARAMETER SETTINGS

To normalize the soft prompts in VLMs, two types of normalization losses are proposed: the Position-Uniform Normalization (PUN) loss and the Position-Aware Normalization (PAN) loss. Both losses involve a crucial hyper-parameter $\omega$, which controls the extent of normalization for soft prompts. Unless specified otherwise, $\omega$ is set to 1 for all datasets, except for the ImageNet dataset where it is set to 10, the OxfordPets dataset, where it is set to 50, and the Food101 dataset where it is also set to 50. Based on our experimental findings, we observed that our approach performs well on these three datasets when $\omega$ is relatively large. This observation aligns with our discovery of a pronounced Low-Norm Effect in these datasets, providing evidence that our method is indeed capable of addressing the Low-Norm Effect. At the same time, we provide a decreasing schedule of $\omega$ for a better balance between $\mathcal{L}_{ce}$ and the PUN or PAN loss. To be specific, it is varied based on a logistic function $\omega_E = 1 - \frac{1}{1+exp(-k(E-0.5\max_E))}$, where $E$ and $\max_E$ denote current training epoch and maximum training epoch, respectively. $k$ represents the attenuation rate, and it is fixed as 0.2.

In addition to $\omega$, the PAN loss incorporates two important hyper-parameters the number of corruption positions $N$ and the pre-defined threshold $\tau$ inducing the Low-Norm Effect. It should be noted that the size of the additional inference cost incurred by the PAN loss is positive correlation with $N$. Technically, the text encoder needs to perform inferences on a batch of $B \times (N + 1)$. In the computational efficiency experiments, we set the default value of $N$ to 1 and $\tau$ to 0.5. These settings are chosen to minimize computational costs while maintaining the desired performance.

Furthermore, the baseline model CoOp has multiple variations, including multiple backbones (e.g. ResNet-50 (He et al., 2016) and ViT-B/16 (Dosovitskiy et al., 2020)), different positions for the class token (e.g., "front", "middle", and "end"), various lengths for the soft-prompt (e.g., 4 and 16), and multiple parameter initialization strategies. For a clear comparison, here we choose one of them as our baseline with ResNet-50 backbone, the class token at the "end" position, 16 soft-prompt tokens, and "random" initialization.

### A.2.3 Corruption Experiments

This subsection provides more details of corruption experiments implemented by the two corruption operations we proposed.

The *REPLACE* operation involves replacing the prompt vector at a single position with a randomly generated vector from a Gaussian distribution, which is characterized by a zero mean and a fixed variance. By modifying the variance of the Gaussian distribution, we can roughly control the norms of generated Gaussian vectors. To be specific, increasing the variance leads to higher norms of the generated Gaussian vectors while decreasing the variance results in lower norms. We employ five different variance values: 0, 0.001, 0.01, 0.1, and 0.5.

Furthermore, the *RESCALE* operation is utilized to rescale the prompt vector at a single position by applying various sizes of rescaling factors, including 0.001, 0.01, 0.1, 0.5, and 2. The first four rescaling factors are employed to reduce the norms, while the last one is utilized to increase the norms. As a result, for soft prompts of length $L$ and a corruption operation, we conduct corruption experiments $L$ times. Then, we can calculate the occurrence frequency with which the performance of corrupted prompts exceeds that of their original counterparts. Tables A2 and A3 provide a detailed record of this occurrence frequency under different corruption operations, respectively. It is noteworthy that the results of occurrence frequency of the Low-Norm Effect across 11 datasets (i.e. the results of Figure 1) are calculated by the sum of four rescaling factors, including 0.001, 0.01, 0.1, and 0.5.

For a better intuitive understanding, we present a part of corruption experiments in the form of bar graphs. Take the ImageNet dataset under 1 shot setting as an example, as shown in Figure A1. From Figure A1(c), we can perceive that the replacement of prompt vectors at positions 1-4 with random Gaussian vector having a zero mean and a variance of 0.001 barely changes the model's performance. Surprisingly, an improvement in performance is observed for positions 5, 7, 8, and 9. Additionally, a similar pattern can be found in Figure A1(h), where we observe varying degrees of performance improvement when the prompt vector at positions 1-14 is rescaled to half of its original magnitude (i.e. reducing the norms). On the contrary, the model's performance for all positions experiences varying degrees of decline when the norms of soft prompts increase, whether the *Replace* or *Rescale* operation, as shown in Figure A1(g), A1(i) and A1(j).

### A.2.4 Base-to-New Results

In this subsection, we compare the baseline model CoCoOp and CoCoOp+Nemesis (ours) in the base-to-new setting. CoCoOp inputs a batch of image features into an additional neural network to generate instance-conditional prompts, which are added to soft prompts to produce a batch of final prompts. Following CoCoOp, all methods are implemented with ViT-B/16 backbone and evaluated with 16 shots. We report the performance on 11 datasets, including the base classes (Base), new classes (New), and the harmonic mean (H) of both. We present comprehensive results of base-to-new experiments conducted on all datasets, as illustrated in Table A4. We can observe that increasing the strength of normalization of soft prompts (i.e. larger $\omega$) would have a slight negative effect on the performance of base classes but better enhance the performance of new classes. This suggests that the generalization performance from base classes to unseen classes can be improved by normalizing the soft prompts of VLMs. In particular, CoCoOp+Nemesis (PAN, $\omega = 20$) achieve a performance improvement of 1.3% compared to CoCoOp for new classes.

### A.2.5 Results of Ablation Study and Hyper-parameter Analysis

This subsection presents the ablation results for the normalized strength $\omega$. The outcomes of different sizes of $\omega$ on few-shot recognition tasks across 11 datasets are illustrated in Table A5. Additionally, Table A7 showcases the results of using different norm types for the PUN loss, including 1-norm, 2-norm, and Inf-norm.

### A.2.6 Results during Training Process

This subsection provides results during soft-prompt tuning VLMs, including training loss, test accuracy, norms of soft prompts at various positions, and the occurrence frequency of Low-Norm Effect for different prompting positions. The data comparison between Figure A2 and Figure A3

leads to several conclusions. Firstly, regardless of whether it is in the 1-shot or 16-shot setting, the PUN loss demonstrates a stronger level of normalization compared to the PAN loss. On the other hand, despite the stronger normalization of the PUN loss, it does not outperform the PAN loss in the 16-shot setting. This implies that simply reducing the norm of soft prompts does not always lead to an improvement in model performance. Secondly, there is a notable occurrence frequency of the Low-Norm Effect in the 16-shot setting during training process, indicating that an increased number of training samples can facilitate the identification of prompting positions that induce the Low-Norm Effect by the PAN loss. This observation may explain why the PAN loss performs better in the 16-shot setting but is inferior to the PUN loss in the 1-shot setting. Lastly, the addition of two normalization losses does not seem to have a substantial influence on the original cross-entropy loss, indicating that they do not contribute to model performance primarily by reducing the cross-entropy loss. Instead, their benefits are likely derived from harnessing the Low-Norm Effect.

### A.2.7 Computation Cost Analysis

The data in Table A8 represents the average computation costs of multiple training batches, based on the benchmark of CoOp's training time. The numbers in parentheses indicate the corresponding running times in seconds. Firstly, we observe that incorporating the PUN or PAN loss increases computation costs compared to using CoOp alone. The PAN loss introduces an additional pre-inference step before each training batch, resulting in a higher computational burden compared to the PUN loss. However, combining both losses does not significantly impact the running time compared to using the PAN loss alone. Furthermore, there is a clear and consistent trend across all methods, including CoOp, PLOT, and our two losses, where computation costs increase with the number of classes in the datasets. We speculate this is caused by the fact that CoOp-based methods optimize the similarity scores between text features of all categories and image features of a mini-batch. The increase of number of classes will result in a significant increase in the gradients required for calculation. Lastly, it should be noted that the running time of the PAN loss and PLOT is almost comparable.

The data in Table A9 demonstrate that increasing the value of $N$ leads to increased computational costs, particularly when dealing with large datasets, resulting in longer computation time and higher memory consumption. This is because the model requires generating a greater number of corrupted textual features for prediction within each training batch. Although it is generally observed that increasing $N$ has a positive impact on the model's performance, finding ways to mitigate the computational burden is a valuable area for future research.

### A.2.8 Other Applicable Scenarios Analysis

While our proposed method primarily focuses on benchmarking soft prompt-tuning VLMs, it should be noted that the benefits of Nemesis may be not limited to it alone. We speculate that they can also be applied to other parameter-efficient tuning (PEFT) methods, such as visual prompt-tuning (Jia et al., 2022) and prefix-tuning (Li & Liang, 2021), as well as their various downstream task.

To verify this, we conducted preliminary experiments on a few PEFT methods and their applicable scenarios, including visual prompt-tuning (Jia et al., 2022) for image classification and prefix-tuning (Li & Liang, 2021) for paraphrase classification. Given that the proposed PAN loss may involve designing specific corruption experiments and adopting distinct performance comparison metrics depending on different methods and tasks. To be more specific, we should adopt various performance metrics when identifying the positions that induce the Low-Norm Effect, such as classification accuracy for classification tasks, mAP (mean Average Precision) for object detection tasks, IoU (Intersection over Union) for segmentation tasks, and BLEU (BilinguaL Evaluation Understudy) for text generation. We leave it for the future work. Here, we only incorporate the PUN loss into these methods and obtain the results, as shown in Tables A10 and A11.

Based on the presented results, the proposed PUN loss can enhance the performance in all conducted experiments. However, determining the ideal weight for the proposed loss requires further investigation, which is a potential direction for future work. In summary, these preliminary results support our research prospects discussed in the Conclusion Section (i.e. Section 6), and we hope that our findings and approaches will provide new insights into PEFT methods and inspire future research in these fields.

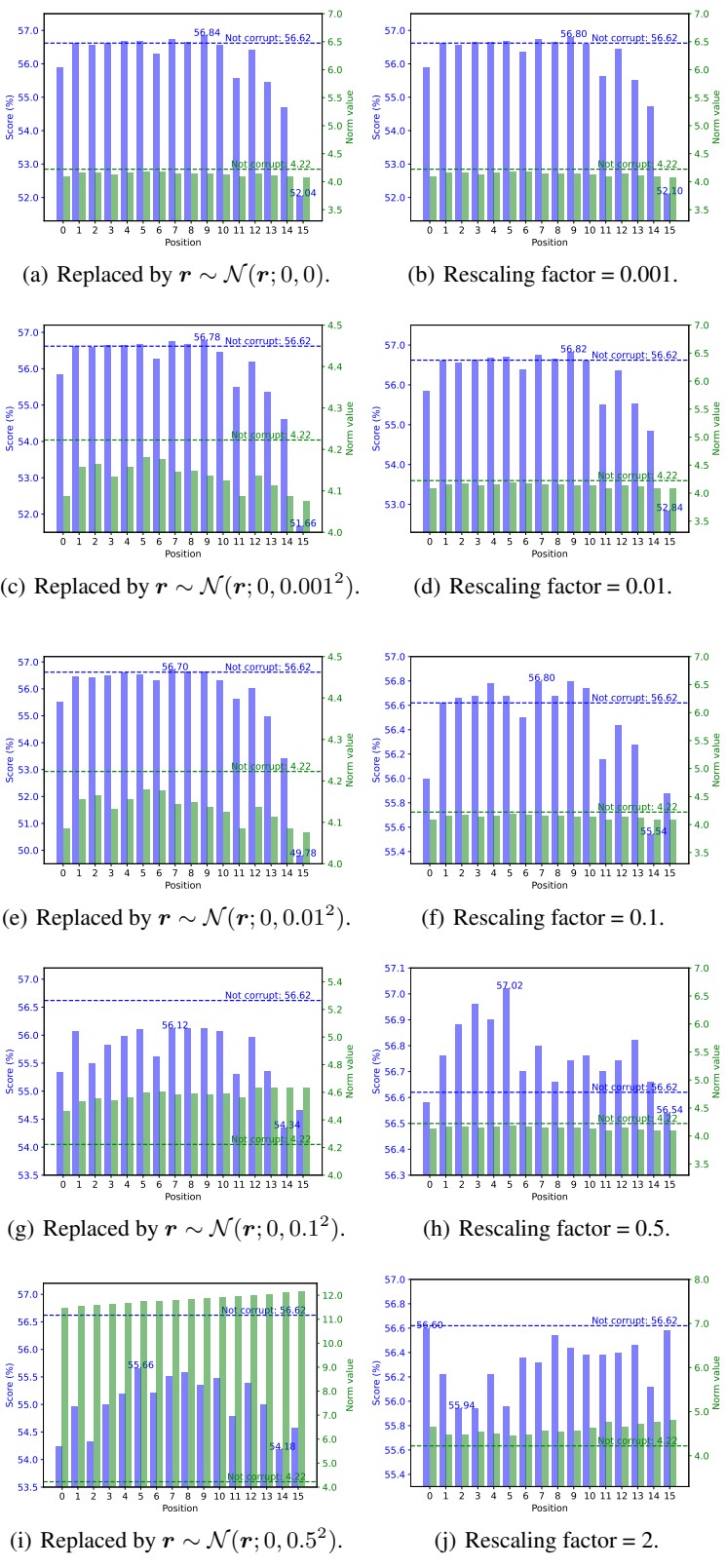

Figure A1: Comparison of the model's performance and norms of soft prompts before and after corrupting the prompt vector at various positions. The blue and green bars denote the model's performance and the norms of soft prompts, respectively. The lines represent the benchmarks of without corruption. Take the ImageNet dataset under 1 shot setting as an example.

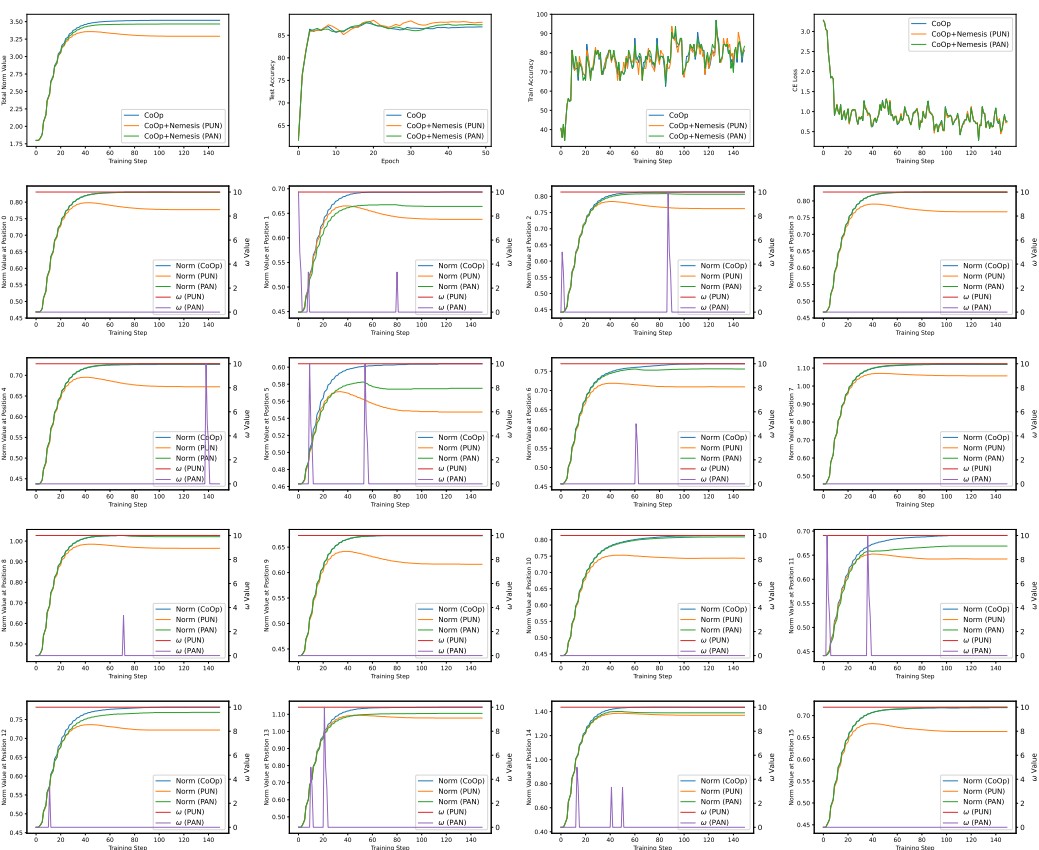

Figure A2: The results of the Caltech101 dataset at the 1-shot setting during the training process.

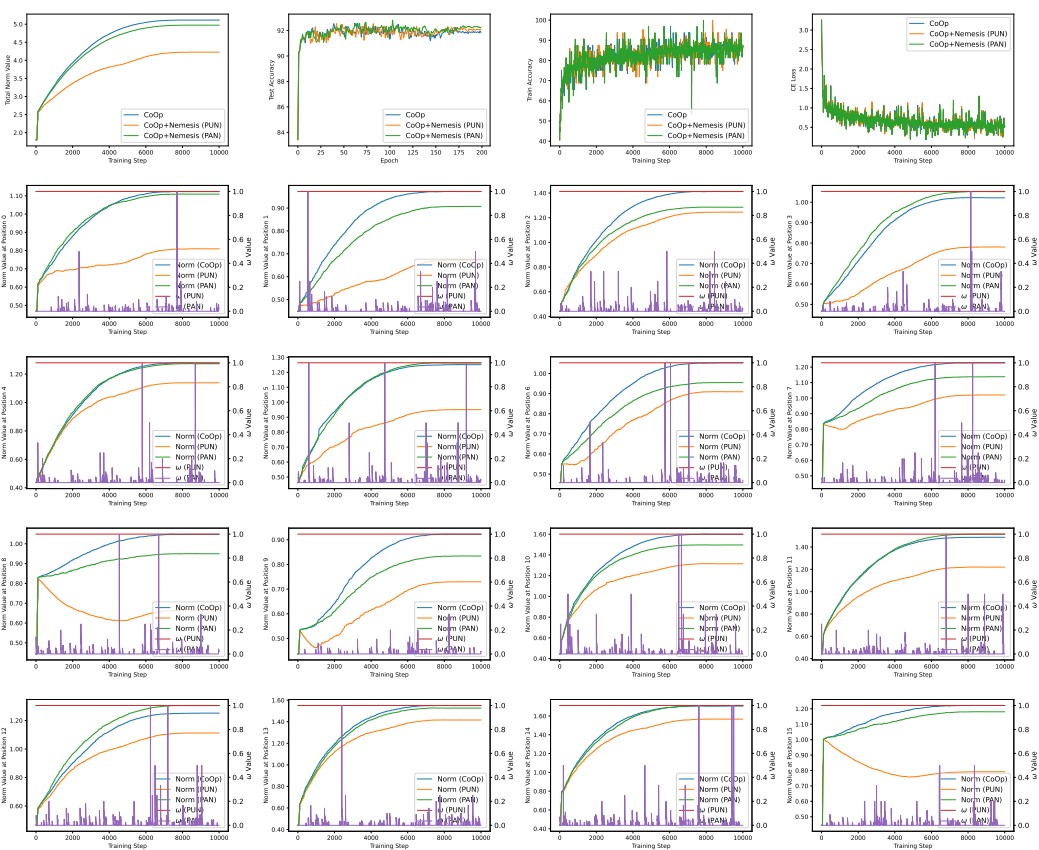

Figure A3: The results of the Caltech101 dataset at the 16-shots setting during the training process.

Table A2: Corruption experiments using the *Replace* operation: the detailed occurrence frequency with which the performance of corrupted prompts exceeds that of their original counterparts. The upward and downward arrows indicate an increase and decrease in norms, respectively.

| Dataset | Variance | 1 shot | 2 shots | 4 shots | 8 shots | 16 shots | Norms variation |
|---|---|---|---|---|---|---|---|
| ImageNet | 0 | 5 | 0 | 0 | 0 | 0 | ↓ |
| | 0.001 | 6 | 0 | 0 | 0 | 0 | ↓ |
| | 0.01 | 3 | 0 | 0 | 0 | 0 | ↓ |
| | 0.1 | 0 | 0 | 0 | 0 | 0 | ↑ |
| | 0.5 | 0 | 0 | 0 | 0 | 0 | ↑ |
| Caltech101 | 0 | 8 | 3 | 0 | 0 | 0 | ↓ |
| | 0.001 | 7 | 4 | 0 | 0 | 0 | ↓ |
| | 0.01 | 7 | 1 | 0 | 0 | 0 | ↓ |
| | 0.1 | 0 | 0 | 0 | 0 | 0 | ↑ |
| | 0.5 | 0 | 0 | 0 | 0 | 0 | ↑ |
| OxfordPets | 0 | 10 | 12 | 7 | 8 | 6 | ↓ |
| | 0.001 | 7 | 12 | 6 | 8 | 5 | ↓ |
| | 0.01 | 6 | 12 | 5 | 4 | 3 | ↓ |
| | 0.1 | 0 | 8 | 0 | 0 | 0 | ↑ |
| | 0.5 | 0 | 3 | 0 | 0 | 0 | ↑ |
| StanfordCars | 0 | 4 | 0 | 0 | 0 | 0 | ↓ |
| | 0.001 | 2 | 0 | 0 | 0 | 0 | ↓ |
| | 0.01 | 3 | 0 | 0 | 0 | 0 | ↓ |
| | 0.1 | 2 | 0 | 0 | 0 | 0 | ↑ |
| | 0.5 | 0 | 0 | 0 | 0 | 0 | ↑ |
| Flowers102 | 0 | 0 | 0 | 0 | 0 | 0 | ↓ |
| | 0.001 | 0 | 0 | 0 | 0 | 0 | ↓ |
| | 0.01 | 0 | 0 | 0 | 0 | 0 | ↓ |
| | 0.1 | 0 | 0 | 0 | 0 | 0 | ↑ |
| | 0.5 | 0 | 0 | 0 | 0 | 0 | ↑ |
| Food101 | 0 | 7 | 5 | 5 | 7 | 1 | ↓ |
| | 0.001 | 7 | 5 | 6 | 7 | 1 | ↓ |
| | 0.01 | 9 | 3 | 5 | 5 | 1 | ↓ |
| | 0.1 | 4 | 3 | 3 | 0 | 0 | ↑ |
| | 0.5 | 1 | 0 | 0 | 0 | 0 | ↑ |
| FGVCAircraft | 0 | 2 | 0 | 0 | 0 | 0 | ↓ |
| | 0.001 | 2 | 0 | 0 | 0 | 0 | ↓ |
| | 0.01 | 2 | 0 | 0 | 0 | 0 | ↓ |
| | 0.1 | 0 | 0 | 0 | 0 | 0 | ↑ |
| | 0.5 | 0 | 0 | 0 | 0 | 0 | ↑ |
| SUN397 | 0 | 0 | 0 | 0 | 0 | 0 | ↓ |
| | 0.001 | 0 | 0 | 0 | 0 | 0 | ↓ |
| | 0.01 | 0 | 0 | 0 | 0 | 0 | ↓ |
| | 0.1 | 0 | 0 | 0 | 0 | 0 | ↑ |
| | 0.5 | 0 | 0 | 0 | 0 | 0 | ↑ |
| DTD | 0 | 3 | 1 | 0 | 0 | 0 | ↓ |
| | 0.001 | 3 | 0 | 0 | 0 | 0 | ↓ |
| | 0.01 | 1 | 0 | 0 | 0 | 0 | ↓ |
| | 0.1 | 0 | 0 | 0 | 0 | 0 | ↑ |
| | 0.5 | 0 | 0 | 0 | 0 | 0 | ↑ |
| EuroSAT | 0 | 0 | 0 | 0 | 0 | 0 | ↓ |
| | 0.001 | 0 | 0 | 0 | 0 | 0 | ↓ |
| | 0.01 | 1 | 0 | 0 | 0 | 0 | ↓ |
| | 0.1 | 1 | 0 | 0 | 0 | 0 | ↑ |
| | 0.5 | 0 | 0 | 0 | 0 | 0 | ↑ |
| UCF101 | 0 | 1 | 0 | 0 | 0 | 0 | ↓ |
| | 0.001 | 1 | 0 | 0 | 0 | 0 | ↓ |
| | 0.01 | 0 | 0 | 0 | 0 | 0 | ↓ |
| | 0.1 | 0 | 0 | 0 | 0 | 0 | ↑ |
| | 0.5 | 0 | 0 | 0 | 0 | 0 | ↑ |

Table A3: Corruption experiments using the *Rescale* operation: the detailed occurrence frequency with which the performance of corrupted prompts exceeds that of their original counterparts. The upward and downward arrows indicate an increase and decrease in norms, respectively.

| Dataset | Resclaing factor | 1 shot | 2 shots | 4 shots | 8 shots | 16 shots | Norms variation |
|---|---|---|---|---|---|---|---|
| ImageNet | 0.001 | 6 | 0 | 0 | 0 | 0 | ↓ |
|  | 0.01 | 6 | 0 | 0 | 0 | 0 | ↓ |
|  | 0.1 | 8 | 3 | 0 | 0 | 0 | ↓ |
|  | 0.5 | 14 | 15 | 13 | 11 | 9 | ↓ |
|  | 2.0 | 0 | 0 | 0 | 1 | 0 | ↑ |
| Caltech101 | 0.001 | 8 | 3 | 0 | 0 | 0 | ↓ |
|  | 0.01 | 9 | 4 | 0 | 0 | 0 | ↓ |
|  | 0.1 | 5 | 5 | 0 | 0 | 0 | ↓ |
|  | 0.5 | 10 | 8 | 11 | 4 | 2 | ↓ |
|  | 2.0 | 3 | 2 | 2 | 1 | 0 | ↑ |
| OxfordPets | 0.001 | 10 | 12 | 5 | 8 | 6 | ↓ |
|  | 0.01 | 10 | 13 | 6 | 7 | 6 | ↓ |
|  | 0.1 | 10 | 13 | 7 | 12 | 6 | ↓ |
|  | 0.5 | 10 | 15 | 16 | 15 | 7 | ↓ |
|  | 2.0 | 1 | 0 | 1 | 0 | 1 | ↑ |
| StanfordCars | 0.001 | 6 | 0 | 0 | 0 | 0 | ↓ |
|  | 0.01 | 5 | 0 | 0 | 0 | 0 | ↓ |
|  | 0.1 | 8 | 0 | 0 | 0 | 0 | ↓ |
|  | 0.5 | 16 | 5 | 4 | 0 | 0 | ↓ |
|  | 2.0 | 6 | 1 | 0 | 0 | 0 | ↑ |
| Flowers102 | 0.001 | 0 | 0 | 0 | 0 | 0 | ↓ |
|  | 0.01 | 0 | 0 | 0 | 0 | 0 | ↓ |
|  | 0.1 | 0 | 0 | 0 | 0 | 0 | ↓ |
|  | 0.5 | 2 | 1 | 0 | 1 | 1 | ↓ |
|  | 2.0 | 0 | 0 | 0 | 0 | 0 | ↑ |
| Food101 | 0.001 | 8 | 4 | 5 | 7 | 1 | ↓ |
|  | 0.01 | 8 | 5 | 6 | 7 | 1 | ↓ |
|  | 0.1 | 13 | 9 | 7 | 7 | 1 | ↓ |
|  | 0.5 | 14 | 12 | 11 | 15 | 13 | ↓ |
|  | 2.0 | 1 | 1 | 1 | 0 | 0 | ↑ |
| FGVCAircraft | 0.001 | 2 | 0 | 0 | 0 | 0 | ↓ |
|  | 0.01 | 1 | 0 | 0 | 0 | 0 | ↓ |
|  | 0.1 | 1 | 0 | 0 | 0 | 0 | ↓ |
|  | 0.5 | 5 | 0 | 0 | 4 | 0 | ↓ |
|  | 2.0 | 3 | 0 | 0 | 0 | 0 | ↑ |
| SUN397 | 0.001 | 0 | 0 | 0 | 0 | 0 | ↓ |
|  | 0.01 | 0 | 0 | 0 | 0 | 0 | ↓ |
|  | 0.1 | 0 | 0 | 0 | 0 | 0 | ↓ |
|  | 0.5 | 8 | 12 | 11 | 5 | 5 | ↓ |
|  | 2.0 | 0 | 0 | 0 | 0 | 0 | ↑ |
| DTD | 0.001 | 3 | 1 | 0 | 0 | 0 | ↓ |
|  | 0.01 | 2 | 1 | 0 | 0 | 0 | ↓ |
|  | 0.1 | 5 | 1 | 0 | 0 | 0 | ↓ |
|  | 0.5 | 13 | 11 | 4 | 2 | 2 | ↓ |
|  | 2.0 | 2 | 1 | 0 | 0 | 0 | ↑ |
| EuroSAT | 0.001 | 0 | 0 | 0 | 0 | 0 | ↓ |
|  | 0.01 | 0 | 0 | 0 | 0 | 0 | ↓ |
|  | 0.1 | 0 | 0 | 0 | 0 | 0 | ↓ |
|  | 0.5 | 8 | 5 | 2 | 2 | 2 | ↓ |
|  | 2.0 | 1 | 5 | 1 | 0 | 2 | ↑ |
| UCF101 | 0.001 | 1 | 0 | 0 | 0 | 0 | ↓ |
|  | 0.01 | 1 | 0 | 0 | 0 | 0 | ↓ |
|  | 0.1 | 1 | 0 | 0 | 0 | 0 | ↓ |
|  | 0.5 | 3 | 11 | 12 | 0 | 2 | ↓ |
|  | 2.0 | 2 | 0 | 1 | 0 | 0 | ↑ |

Table A4: Comparison of CoCoOp and CoCoOp+Nemesis (ours) in the base-to-new generalization setting. Following CoCoOp, all methods are implemented with ViT-B/16 backbone and evaluated with 16 shots. We report the performance on 11 datasets, including the base classes (Base), new classes (New), and the harmonic mean (H) of both.

(a) **Average**.

|  | Base | New | H |
|---|---|---|---|
| CoCoOp | 80.8 | 72.6 | 76.5 |
| CoCoOp+Nemesis (PUN, $\omega = 1$) | 80.7 | 72.6 | 76.4 |
| CoCoOp+Nemesis (PUN, $\omega = 10$) | 80.4 | 73.2 | 76.6 |
| CoCoOp+Nemesis (PUN, $\omega = 20$) | 80.5 | 73.3 | 76.7 |
| CoCoOp+Nemesis (PAN, $\omega = 1$) | 80.4 | 72.6 | 76.3 |
| CoCoOp+Nemesis (PAN, $\omega = 10$) | 80.5 | 73.3 | 76.7 |
| CoCoOp+Nemesis (PAN, $\omega = 20$) | 80.3 | 73.9 | 77.0 |

(b) ImageNet.

|  | Base | New | H |
|---|---|---|---|
| CoCoOp | 76.3 | 70.6 | 73.3 |
| CoCoOp+Nemesis (PUN, $\omega = 1$) | 76.2 | 70.7 | 73.3 |
| CoCoOp+Nemesis (PUN, $\omega = 10$) | 76.2 | 70.6 | 73.3 |
| CoCoOp+Nemesis (PUN, $\omega = 20$) | 76.3 | 70.5 | 73.3 |
| CoCoOp+Nemesis (PAN, $\omega = 1$) | 76.2 | 70.5 | 73.2 |
| CoCoOp+Nemesis (PAN, $\omega = 10$) | 76.2 | 70.5 | 73.3 |
| CoCoOp+Nemesis (PAN, $\omega = 20$) | 76.2 | 70.5 | 73.2 |

(c) Caltech101.

|  | Base | New | H |
|---|---|---|---|
| CoCoOp | 97.9 | 93.2 | 95.5 |
| CoCoOp+Nemesis (PUN, $\omega = 1$) | 97.8 | 93.6 | 95.7 |
| CoCoOp+Nemesis (PUN, $\omega = 10$) | 97.7 | 93.7 | 95.6 |
| CoCoOp+Nemesis (PUN, $\omega = 20$) | 98.0 | 93.7 | 95.8 |
| CoCoOp+Nemesis (PAN, $\omega = 1$) | 97.8 | 93.5 | 95.6 |
| CoCoOp+Nemesis (PAN, $\omega = 10$) | 97.8 | 93.9 | 95.8 |
| CoCoOp+Nemesis (PAN, $\omega = 20$) | 98.0 | 93.7 | 95.8 |

(d) OxfordPets.

|  | Base | New | H |
|---|---|---|---|
| CoCoOp | 95.0 | 97.6 | 96.3 |
| CoCoOp+Nemesis (PUN, $\omega = 1$) | 95.2 | 97.8 | 96.5 |
| CoCoOp+Nemesis (PUN, $\omega = 10$) | 95.0 | 97.6 | 96.3 |
| CoCoOp+Nemesis (PUN, $\omega = 20$) | 94.8 | 97.7 | 96.2 |
| CoCoOp+Nemesis (PAN, $\omega = 1$) | 95.3 | 97.7 | 96.5 |
| CoCoOp+Nemesis (PAN, $\omega = 10$) | 94.8 | 97.8 | 96.3 |
| CoCoOp+Nemesis (PAN, $\omega = 20$) | 94.7 | 97.7 | 96.2 |

(e) StanfordCars.

|  | Base | New | H |
|---|---|---|---|
| CoCoOp | 70.8 | 73.1 | 71.9 |
| CoCoOp+Nemesis (PUN, $\omega = 1$) | 71.0 | 73.3 | 72.1 |
| CoCoOp+Nemesis (PUN, $\omega = 10$) | 70.2 | 73.6 | 71.9 |
| CoCoOp+Nemesis (PUN, $\omega = 20$) | 70.6 | 73.4 | 72.0 |
| CoCoOp+Nemesis (PAN, $\omega = 1$) | 71.0 | 72.7 | 71.5 |
| CoCoOp+Nemesis (PAN, $\omega = 10$) | 70.2 | 73.5 | 72.1 |
| CoCoOp+Nemesis (PAN, $\omega = 20$) | 70.6 | 73.2 | 72.3 |

(f) Flowers102.

|  | Base | New | H |
|---|---|---|---|
| CoCoOp | 95.2 | 68.9 | 79.9 |
| CoCoOp+Nemesis (PUN, $\omega = 1$) | 95.0 | 70.8 | 81.1 |
| CoCoOp+Nemesis (PUN, $\omega = 10$) | 93.7 | 71.3 | 81.0 |
| CoCoOp+Nemesis (PUN, $\omega = 20$) | 94.0 | 71.3 | 81.1 |
| CoCoOp+Nemesis (PAN, $\omega = 1$) | 94.5 | 72.2 | 81.8 |
| CoCoOp+Nemesis (PAN, $\omega = 10$) | 95.2 | 71.8 | 81.8 |
| CoCoOp+Nemesis (PAN, $\omega = 20$) | 93.5 | 71.5 | 81.0 |

(g) Food101.

|  | Base | New | H |
|---|---|---|---|
| CoCoOp | 90.5 | 91.3 | 90.9 |
| CoCoOp+Nemesis (PUN, $\omega = 1$) | 90.6 | 91.5 | 91.0 |
| CoCoOp+Nemesis (PUN, $\omega = 10$) | 90.6 | 91.4 | 91.0 |
| CoCoOp+Nemesis (PUN, $\omega = 20$) | 90.6 | 91.5 | 91.0 |
| CoCoOp+Nemesis (PAN, $\omega = 1$) | 90.5 | 91.5 | 91.0 |
| CoCoOp+Nemesis (PAN, $\omega = 10$) | 90.5 | 91.3 | 90.9 |
| CoCoOp+Nemesis (PAN, $\omega = 20$) | 90.6 | 91.5 | 91.0 |

(h) FGVCAircraft.

|  | Base | New | H |
|---|---|---|---|
| CoCoOp | 35.6 | 32.1 | 33.8 |
| CoCoOp+Nemesis (PUN, $\omega = 1$) | 35.7 | 32.7 | 34.1 |
| CoCoOp+Nemesis (PUN, $\omega = 10$) | 35.5 | 34.0 | 34.7 |
| CoCoOp+Nemesis (PUN, $\omega = 20$) | 35.8 | 34.8 | 35.3 |
| CoCoOp+Nemesis (PAN, $\omega = 1$) | 35.3 | 33.4 | 34.3 |
| CoCoOp+Nemesis (PAN, $\omega = 10$) | 35.5 | 32.3 | 33.8 |
| CoCoOp+Nemesis (PAN, $\omega = 20$) | 35.1 | 35.8 | 35.4 |

(i) SUN397.

|  | Base | New | H |
|---|---|---|---|
| CoCoOp | 79.6 | 76.6 | 78.1 |
| CoCoOp+Nemesis (PUN, $\omega = 1$) | 79.2 | 75.9 | 77.5 |
| CoCoOp+Nemesis (PUN, $\omega = 10$) | 79.2 | 76.7 | 77.9 |
| CoCoOp+Nemesis (PUN, $\omega = 20$) | 79.4 | 77.0 | 78.2 |
| CoCoOp+Nemesis (PAN, $\omega = 1$) | 79.3 | 76.7 | 78.0 |
| CoCoOp+Nemesis (PAN, $\omega = 10$) | 79.1 | 77.0 | 78.0 |
| CoCoOp+Nemesis (PAN, $\omega = 20$) | 79.2 | 76.7 | 77.9 |

(j) DTD.

|  | Base | New | H |
|---|---|---|---|
| CoCoOp | 77.3 | 55.8 | 64.8 |
| CoCoOp+Nemesis (PUN, $\omega = 1$) | 77.5 | 55.4 | 64.6 |
| CoCoOp+Nemesis (PUN, $\omega = 10$) | 77.4 | 54.9 | 64.2 |
| CoCoOp+Nemesis (PUN, $\omega = 20$) | 76.4 | 58.7 | 66.4 |
| CoCoOp+Nemesis (PAN, $\omega = 1$) | 77.3 | 56.6 | 65.3 |
| CoCoOp+Nemesis (PAN, $\omega = 10$) | 76.0 | 57.3 | 65.3 |
| CoCoOp+Nemesis (PAN, $\omega = 20$) | 76.6 | 56.7 | 65.2 |

(k) EuroSAT.

|  | Base | New | H |
|---|---|---|---|
| CoCoOp | 88.1 | 65.2 | 74.9 |
| CoCoOp+Nemesis (PUN, $\omega = 1$) | 87.4 | 64.1 | 74.0 |
| CoCoOp+Nemesis (PUN, $\omega = 10$) | 86.6 | 66.9 | 75.5 |
| CoCoOp+Nemesis (PUN, $\omega = 20$) | 87.5 | 64.0 | 73.9 |
| CoCoOp+Nemesis (PAN, $\omega = 1$) | 85.8 | 61.8 | 71.9 |
| CoCoOp+Nemesis (PAN, $\omega = 10$) | 88.0 | 68.6 | 77.1 |
| CoCoOp+Nemesis (PAN, $\omega = 20$) | 86.2 | 71.3 | 78.1 |

(l) UCF101.

|  | Base | New | H |
|---|---|---|---|
| CoCoOp | 82.5 | 73.8 | 77.9 |
| CoCoOp+Nemesis (PUN, $\omega = 1$) | 81.9 | 72.6 | 77.0 |
| CoCoOp+Nemesis (PUN, $\omega = 10$) | 81.9 | 74.4 | 78.0 |
| CoCoOp+Nemesis (PUN, $\omega = 20$) | 81.9 | 74.1 | 77.8 |
| CoCoOp+Nemesis (PAN, $\omega = 1$) | 82.1 | 72.5 | 77.0 |
| CoCoOp+Nemesis (PAN, $\omega = 10$) | 81.7 | 72.6 | 76.9 |
| CoCoOp+Nemesis (PAN, $\omega = 20$) | 81.6 | 74.5 | 77.9 |

Table A5: Comparison of CoOp and CoOp+Nemesis (ours) in few-shot image recognition tasks, including 11 different datasets. All methods are implemented with ResNet-50 backbone.

| Dataset | Methods | 1 shot | 2 shots | 4 shots | 8 shots | 16 shots |
|---------|---------|--------|---------|---------|---------|----------|
| ImageNet | CoOp | 56.62 | 56.96 | 59.40 | 61.38 | 62.70 |
| | Nemesis (PUN, $\omega$=0.1) | 56.62 | 57.12 | 59.54 | 61.58 | 63.18 |
| | Nemesis (PUN, $\omega$=1) | 56.16 | 58.00 | 61.14 | 62.30 | 62.96 |
| | Nemesis (PUN, $\omega$=10) | **60.44** | **61.78** | **62.14** | **62.54** | 63.08 |
| | Nemesis (PAN, $\omega$=0.1) | 57.00 | 57.44 | 59.64 | 61.76 | 63.14 |
| | Nemesis (PAN, $\omega$=1) | 56.70 | 57.10 | 59.54 | 61.78 | **63.28** |
| | Nemesis (PAN, $\omega$=10) | 57.08 | 58.52 | 61.14 | 62.18 | 63.14 |
| Caltech101 | CoOp | 86.98 | 87.32 | 89.28 | 89.82 | 91.70 |
| | Nemesis (PUN, $\omega$=0.1) | **87.64** | 87.36 | 89.46 | 90.32 | 91.86 |
| | Nemesis (PUN, $\omega$=1) | 86.90 | 86.98 | 88.94 | 90.32 | 91.78 |
| | Nemesis (PUN, $\omega$=10) | 85.76 | **87.92** | **90.36** | **91.28** | 91.42 |
| | Nemesis (PAN, $\omega$=0.1) | 87.36 | 87.50 | 89.50 | 90.36 | 91.70 |
| | Nemesis (PAN, $\omega$=1) | 87.58 | 87.44 | 89.50 | 90.12 | **91.90** |
| | Nemesis (PAN, $\omega$=10) | 87.32 | 87.06 | 88.92 | 90.16 | 91.70 |
| OxfordPets | CoOp | 85.80 | 83.02 | 85.98 | 84.86 | 85.98 |
| | Nemesis (PUN, $\omega$=0.1) | **86.28** | 83.16 | 86.38 | 85.20 | 86.18 |
| | Nemesis (PUN, $\omega$=1) | 86.22 | 82.68 | 85.82 | 83.76 | 86.08 |
| | Nemesis (PUN, $\omega$=50) | 85.02 | **87.18** | **88.00** | **88.66** | 88.88 |
| | Nemesis (PAN, $\omega$=0.1) | **86.28** | 83.16 | 86.38 | 85.20 | 86.18 |
| | Nemesis (PAN, $\omega$=1) | 86.22 | 82.68 | 85.82 | 83.76 | 86.08 |
| | Nemesis (PAN, $\omega$=50) | 85.90 | 83.48 | 86.90 | 87.68 | **88.90** |
| StanfordCars | CoOp | 55.62 | 58.44 | 62.84 | 67.78 | 72.80 |
| | Nemesis (PUN, $\omega$=0.1) | 56.18 | 58.50 | 63.00 | 68.22 | **73.74** |
| | Nemesis (PUN, $\omega$=1) | 56.16 | 58.08 | **63.72** | 68.10 | 71.50 |
| | Nemesis (PUN, $\omega$=10) | **56.42** | **59.62** | 63.34 | 67.08 | 70.46 |
| | Nemesis (PAN, $\omega$=0.1) | 56.20 | 58.70 | 63.02 | 68.24 | 73.56 |
| | Nemesis (PAN, $\omega$=1) | 56.14 | 58.54 | 63.46 | **68.52** | 73.60 |
| | Nemesis (PAN, $\omega$=10) | 56.16 | 58.08 | 63.64 | 67.66 | 71.60 |
| Flowers102 | CoOp | 67.48 | 76.92 | 85.28 | 90.84 | 94.30 |
| | Nemesis (PUN, $\omega$=0.1) | 67.70 | 77.04 | 85.46 | 91.68 | 94.76 |
| | Nemesis (PUN, $\omega$=1) | 68.52 | 78.38 | 86.58 | **92.08** | 94.62 |
| | Nemesis (PUN, $\omega$=10) | **69.96** | **80.16** | 86.56 | 90.46 | 93.70 |
| | Nemesis (PAN, $\omega$=0.1) | 67.76 | 77.44 | 85.86 | 91.50 | 94.58 |
| | Nemesis (PAN, $\omega$=1) | 67.94 | 77.12 | 85.94 | 91.80 | **94.90** |
| | Nemesis (PAN, $\omega$=10) | 68.44 | 78.86 | **87.16** | 91.92 | 94.46 |
| Food101 | CoOp | 73.76 | 72.62 | 74.06 | 71.80 | 74.20 |
| | Nemesis (PUN, $\omega$=0.1) | 74.02 | 72.84 | 74.12 | 71.68 | 74.32 |
| | Nemesis (PUN, $\omega$=1) | 73.70 | 71.18 | 71.34 | 71.44 | 76.24 |
| | Nemesis (PUN, $\omega$=50) | **74.26** | **77.36** | **78.68** | **78.80** | 78.62 |
| | Nemesis (PAN, $\omega$=0.1) | 74.10 | 72.98 | 74.34 | 72.06 | 74.48 |
| | Nemesis (PAN, $\omega$=1) | 74.08 | 72.84 | 74.06 | 71.52 | 74.28 |
| | Nemesis (PAN, $\omega$=50) | 72.10 | 71.74 | 75.30 | 78.38 | **79.36** |
| FGVCAircraft | CoOp | 9.56 | 18.30 | 21.04 | 26.66 | 31.64 |
| | Nemesis (PUN, $\omega$=0.1) | 9.36 | 19.42 | 21.46 | **27.94** | 31.64 |
| | Nemesis (PUN, $\omega$=1) | 10.02 | 19.40 | 21.76 | 27.04 | 30.92 |
| | Nemesis (PUN, $\omega$=10) | **12.48** | **19.88** | **22.72** | 26.86 | 29.64 |
| | Nemesis (PAN, $\omega$=0.1) | 9.82 | 18.76 | 21.36 | 27.10 | 31.72 |
| | Nemesis (PAN, $\omega$=1) | 9.88 | 18.52 | 20.94 | 27.50 | **32.32** |
| | Nemesis (PAN, $\omega$=10) | 10.28 | 19.36 | 22.16 | 27.48 | 30.50 |
| SUN397 | CoOp | 60.12 | 59.96 | 62.70 | 65.28 | 68.82 |
| | Nemesis (PUN, $\omega$=0.1) | 60.12 | 60.14 | 62.78 | 65.64 | 69.36 |
| | Nemesis (PUN, $\omega$=1) | 59.54 | 58.60 | 63.54 | 66.62 | 69.82 |
| | Nemesis (PUN, $\omega$=10) | 59.16 | **61.38** | 65.08 | 67.30 | 69.74 |
| | Nemesis (PAN, $\omega$=0.1) | **60.32** | 59.96 | 63.06 | 65.74 | 69.42 |
| | Nemesis (PAN, $\omega$=1) | 59.36 | 59.56 | 64.06 | 66.82 | 69.72 |
| | Nemesis (PAN, $\omega$=10) | 58.78 | 59.80 | **64.92** | **67.84** | **69.92** |
| DTD | CoOp | 44.16 | 46.24 | 53.16 | 58.64 | 62.70 |
| | Nemesis (PUN, $\omega$=0.1) | **44.60** | 46.28 | **53.64** | 58.85 | 63.06 |
| | Nemesis (PUN, $\omega$=1) | 43.98 | 46.64 | 53.48 | 58.74 | 63.16 |
| | Nemesis (PUN, $\omega$=10) | 43.96 | 46.20 | 53.46 | 58.98 | 62.76 |
| | Nemesis (PAN, $\omega$=0.1) | 43.96 | **46.94** | 53.10 | 58.96 | 63.06 |
| | Nemesis (PAN, $\omega$=1) | 43.76 | 46.64 | 53.54 | 59.00 | **63.26** |
| | Nemesis (PAN, $\omega$=10) | 44.58 | 46.34 | 53.46 | **59.16** | 63.16 |
| EuroSAT | CoOp | 48.78 | 59.90 | 67.54 | 77.32 | 83.24 |
| | Nemesis (PUN, $\omega$=0.1) | 48.14 | 61.60 | 68.64 | 77.12 | 83.42 |
| | Nemesis (PUN, $\omega$=1) | 49.06 | 60.62 | 69.26 | 77.22 | 83.50 |
| | Nemesis (PUN, $\omega$=10) | **51.10** | 61.22 | **69.52** | 77.10 | 82.86 |
| | Nemesis (PAN, $\omega$=0.1) | 50.12 | **62.08** | 69.20 | **77.70** | 83.10 |
| | Nemesis (PAN, $\omega$=1) | 50.66 | 61.24 | 67.68 | 77.36 | **83.52** |
| | Nemesis (PAN, $\omega$=10) | 50.62 | 61.70 | 67.94 | 77.60 | 83.36 |
| UCF101 | CoOp | 62.66 | 64.40 | 68.50 | 73.46 | 76.58 |
| | Nemesis (PUN, $\omega$=0.1) | **63.38** | 65.00 | 69.26 | 73.28 | 76.58 |
| | Nemesis (PUN, $\omega$=1) | 63.14 | 64.40 | **69.32** | 73.36 | **77.28** |
| | Nemesis (PUN, $\omega$=10) | 60.88 | 64.66 | 69.04 | 72.74 | 75.80 |
| | Nemesis (PAN, $\omega$=0.1) | 63.14 | 64.74 | 68.64 | 73.62 | 76.94 |
| | Nemesis (PAN, $\omega$=1) | 63.18 | **65.24** | 69.14 | 73.52 | 77.10 |
| | Nemesis (PAN, $\omega$=10) | 62.44 | 64.60 | 68.76 | **74.04** | 77.02 |

Table A6: Hyper-parameter analysis on threshold $\tau$.

| Threshold $\tau$ | Caltech101 | StanfordCars | EuroSAT | DTD | Flowers102 | UCF101 | Average |
|---|---|---|---|---|---|---|---|
| 0.1 | 89.29 | 63.91 | 67.79 | 53.32 | 83.35 | 69.52 | 85.44 |
| 0.5* | 89.25 | 63.59 | 67.13 | 53.30 | 83.44 | 69.58 | 85.26 |
| 0.9 | 89.31 | 63.81 | 68.28 | 53.55 | 83.51 | 69.40 | 85.57 |

Table A7: Hyper-parameter analysis on different types of p-norm.

| Norm type | Caltech101 | StanfordCars | EuroSAT | DTD | Flowers102 | UCF101 | Average |
|---|---|---|---|---|---|---|---|
| 2-norm* | 89.25 | 63.59 | 67.13 | 53.30 | 83.44 | 69.58 | 85.26 |
| 1-norm | 89.16 | 63.48 | 67.43 | 53.22 | 84.21 | 68.76 | 85.25 |
| Inf-norm | 89.33 | 63.95 | 67.21 | 53.23 | 83.39 | 69.52 | 85.33 |

Table A8: Analysis of computation costs. All comparison results are obtained by the average of multiple training batches and based on the benchmark of the training time of CoOp. For fair comparison, we do not report the computation costs of PLOT on the ImageNet dataset as it needs to reduce batch size for single GPU execution. The PAN loss adopts the default setting $N = 1$.

| Datasets | Classes | CoOp | CoOp+PUN | CoOp+PAN | CoOp+PUN+PAN | PLOT |
|---|---|---|---|---|---|---|
| ImageNet | 1000 | 1× (0.22) | 2.27× (0.50) | 4.00× (0.88) | 4.05× (0.89) | - |
| SUN397 | 397 | 1× (0.11) | 1.91× (0.21) | 3.36× (0.37) | 3.45× (0.38) | 3.00× (0.33) |
| StanfordCars | 196 | 1× (0.07) | 1.57× (0.11) | 2.86× (0.20) | 2.86× (0.20) | 2.57× (0.18) |
| Flowers102 | 102 | 1× (0.06) | 1.17× (0.07) | 2.00× (0.12) | 2.00× (0.12) | 1.83× (0.11) |
| UCF101 | 101 | 1× (0.06) | 1.17× (0.07) | 2.00× (0.12) | 2.17× (0.13) | 1.83× (0.11) |
| Food101 | 101 | 1× (0.06) | 1.17× (0.07) | 2.00× (0.12) | 2.00× (0.12) | 1.83× (0.11) |
| Caltech101 | 100 | 1× (0.06) | 1.17× (0.07) | 2.00× (0.12) | 2.00× (0.12) | 1.83× (0.11) |
| FGVCAircraft | 100 | 1× (0.06) | 1.17× (0.07) | 2.00× (0.12) | 2.00× (0.12) | 1.83× (0.11) |
| DTD | 47 | 1× (0.05) | 1.00× (0.05) | 1.60× (0.08) | 1.60× (0.08) | 1.40× (0.07) |
| OxfordPets | 37 | 1× (0.05) | 1.00× (0.05) | 1.40× (0.07) | 1.60× (0.08) | 1.40× (0.07) |
| EuroSAT | 10 | 1× (0.05) | 1.00× (0.05) | 1.20× (0.06) | 1.20× (0.06) | 1.00× (0.05) |

Table A9: Analysis of computation costs about $N$. All comparison results are obtained by the average of multiple training batches and based on the benchmark of the training time of CoOp.

| Datasets | CoOp | CoOp+PAN($N=1$) | CoOp+PAN($N=2$) | CoOp+PAN($N=4$) | CoOp+PAN($N=8$) | CoOp+PAN($N=16$) |
|---|---|---|---|---|---|---|
| ImageNet | 1× (0.22) | 4.00× (0.88) | 4.86× (1.07) | 6.64× (1.46) | 10.09× (2.22) | 17.05× (3.75) |
| SUN397 | 1× (0.11) | 3.36× (0.37) | 4.09× (0.45) | 5.36× (0.59) | 8.18× (0.90) | 13.64× (1.50) |
| StanfordCars | 1× (0.07) | 2.86× (0.20) | 3.43× (0.24) | 4.43× (0.31) | 6.57× (0.46) | 10.71× (0.75) |
| Flowers102 | 1× (0.06) | 2.00× (0.12) | 2.50× (0.15) | 3.00× (0.18) | 4.43× (0.26) | 6.83× (0.41) |
| UCF101 | 1× (0.06) | 2.00× (0.12) | 2.33× (0.14) | 3.00× (0.18) | 4.43× (0.26) | 6.83× (0.41) |
| Food101 | 1× (0.06) | 2.00× (0.12) | 2.33× (0.14) | 3.00× (0.18) | 4.43× (0.26) | 6.67× (0.40) |
| Caltech101 | 1× (0.06) | 2.00× (0.12) | 2.33× (0.14) | 3.00× (0.18) | 4.17× (0.25) | 6.67× (0.40) |
| FGVCAircraft | 1× (0.06) | 2.00× (0.12) | 2.33× (0.14) | 3.00× (0.18) | 4.17× (0.25) | 6.67× (0.40) |
| DTD | 1× (0.05) | 1.60× (0.08) | 1.80× (0.09) | 2.20× (0.11) | 2.80× (0.14) | 4.20× (0.21) |
| OxfordPets | 1× (0.05) | 1.40× (0.07) | 1.80× (0.09) | 1.80× (0.09) | 2.40× (0.12) | 3.40× (0.17) |
| EuroSAT | 1× (0.05) | 1.20× (0.06) | 1.20× (0.06) | 1.20× (0.06) | 1.40× (0.07) | 1.80× (0.09) |

Table A10: Results from incorporating the PUN loss into visual prompt tuning for image classification. The experiment setting follows the VTAB-1k (Zhai et al., 2019) benchmark. $\omega = 0$ denotes the case without the PUN loss.

| Datasets | $\omega$=0 | $\omega$=0.001 | $\omega$=0.01 | $\omega$=0.1 | $\omega$=1 | $\omega$=10 |
|---|---|---|---|---|---|---|
| Sun397 (Natural) | 49.86 | 49.76 | **50.04** | 48.89 | 46.67 | 45.07 |
| EuroSAT (Specialized) | 91.89 | 91.30 | 92.00 | **92.81** | 91.63 | 91.31 |
| DMLab (Structured) | 33.54 | 33.39 | 33.75 | 34.85 | **35.75** | 35.50 |

Table A11: Results from incorporating the PUN loss into prefix-tuning for paraphrase classification. The experiment is conducted on the MRPC dataset (Dolan & Brockett, 2005). $\omega = 0$ denotes the case without the PUN loss.

| Datasets | Performance Metrics | $\omega$=0 | $\omega$=0.0001 | $\omega$=0.001 | $\omega$=0.01 | $\omega$=0.1 | $\omega$=1 |
|---|---|---|---|---|---|---|---|
| MRPC | F1 Score | 89.37 | **91.06** | 90.90 | 88.70 | 80.62 | 80.81 |
| | Accuracy | 85.10 | **88.00** | 88.00 | 85.10 | 69.62 | 70.90 |

