# OpenReview forum: "Nemesis: Normalizing the Soft-prompt Vectors of Vision-Language Models"
_ICLR.cc/2024/Conference — ICLR 2024 spotlight_

### Official Review · Reviewer_bzG1 · 2023-10-28

**Soundness:** 3 good
**Presentation:** 3 good
**Contribution:** 3 good
**Rating:** 6
**Confidence:** 3

**Summary:**

This paper answers the question "do we need to normalize the soft prompts in VLMs?" by (1) uncovering a phenomenon called the low-norm effect and (2) proposing a new method named normalizing the soft-prompt vectors of vision-language models (Nemesis) to normalize soft-prompt vectors in VLMs. The contributions include, (1) new soft-prompt vector normalization method for VLMs (normalizing soft prompts during soft-prompt tuning), (2) better results when evaluated by domain generalization settings for VLMs.

**Strengths:**

(1) new soft-prompt vector normalization method for VLMs, which can be incorporated into any soft-prompt based methods;
(2) better results when evaluated by domain generalization settings for VLMs.

**Weaknesses:**

1. prefer to learn more details of how you decide the length of soft prompt vectors, e.g., why 4 and 16, will there be more ranges to be investigated basing on the specificl tasks for VLMs?
2. prefer to learn more investigations of combining Nemesis with existing PEFT algorithms to see if the results can be further improved or not so that other researchers can better leverage your method to their existing frameworks.

**Questions:**

1. could there be a combination of between soft-prompt tuning and hard-prompt tuning? (hard = explicitly use some predefined words/phrases as part of the prompts);
2. any idea of further combining existing PEFT (prompt tuning, prefix tuning, LoRA...) with your Nemesis method?

---

> ### Author Response · Authors · 2023-11-19
> **Reply for Reviewer bzG1**
>
> We greatly thank you for your valuable insights and valuable feedback. We address your concerns as follows.
>
> > Q1. Prefer to learn more details of how you decide the length of soft prompt vectors, e.g., why 4 and 16, will there be more ranges to be investigated basing on the specificl tasks for VLMs?
>
> **A1.** Thanks for your valuable comment regarding the discussion of the impact of the length of soft prompt vectors. Our paper mainly focuses on investigating the effect of soft prompt norms on vision-language models, so we just follow the standard settings of CoOp, with a length of 16 for soft prompts. Generally, the length of soft prompts is typically determined empirically. However, we do have observed better results with certain datasets using different lengths, such as 6 and 10, compared to the standard length of 16, through experiments. This suggests that the standard settings of CoOp may not be optimal for all datasets, indicating the need to explore and determine an appropriate length of soft prompts, especially across different datasets. Therefore, we believe it is **necessary and beneficial** to try more lengths of soft prompts when dealing with various VLM-based tasks.
>
> ----
>
> > Q2. Prefer to learn more investigations of combining Nemesis with existing PEFT algorithms to see if the results can be further improved or not so that other researchers can better leverage your method to their existing frameworks.
>
> **A2.** We appreciate your concern regarding the need for discussion on potential applicable scenarios. While our proposed method primarily focuses on benchmarking soft prompt-tuning VLMs and several downstream VLM-based tasks, including few-shot image classification, domain generalization, and base-to-new generalization, it should be noted that the benefits of the two proposed normalization losses may be not limited to these tasks alone. We speculate that they can also be applied to other parameter-efficient tuning (**PEFT**) methods, such as prompt-tuning, prefix-tuning, P-tuning, adapter-tuning, and LoRA, as well as their various downstream tasks, including vision-related tasks like image classification, object detection and semantic segmentation, as well as language-related tasks like text classification and generation.
>
> To verify this, we conducted preliminary experiments on a few PEFT methods and their applicable scenarios, including **visual prompt-tuning (VPT)** [1] for image classification and **prefix-tuning** [2] for paraphrase classification. Due to time constraints during the rebuttal period, we focused exclusively on testing our proposed PEN loss, which can be seamlessly and effortlessly integrated into existing PEFT methods. On the other hand, the proposed PAN loss may involve designing specific corruption experiments and adopting distinct performance comparison metrics depending on different methods and tasks. To be more specific, we should adopt various performance metrics when identifying the positions that induce the Low-Norm Effect, such as classification accuracy for classification tasks, mAP (mean Average Precision) for object detection tasks, IoU (Intersection over Union) for segmentation tasks, and BLEU (BilinguaL Evaluation Understudy) for text generation. We hope for more research to explore its applicability and adaptability in various PEFT tasks in the future. Hence, here we only report our preliminary results based on the PEN loss using various values of $\omega$, as shown in following tables.  $\omega$ is a scaling coefficient that controls the normalization strength.

---

> ### Author Response · Authors · 2023-11-19
> **Reply for Reviewer bzG1**
>
> | Datasets | $\omega$=0 | $\omega$=0.001 | $\omega$=0.01 | $\omega$=0.1 | $\omega$=1 | $\omega$=10 |
> | :---- | :----: | :----: | :----: | :----: | :----: | :----: |
> | Sun397 (Natural) | 49.86 | 49.76 | **50.04** | 48.89 | 46.67 | 45.07 |
> | EuroSAT (Specialized) | 91.89 | 91.30 | 92.00 | **92.81** | 91.63 | 91.31 |
> | DMLab (Structured) | 33.54 | 33.39 | 33.75 | 34.85 | **35.75** | 35.50 |
>
> | Datasets | Performance Metrics | $\omega$=0 | $\omega$=0.0001 | $\omega$=0.001 | $\omega$=0.01 | $\omega$=0.1 | $\omega$=1 |
> | :---- | :----: | :----: | :----: | :----: | :----: | :----: | :----: |
> | MRPC | F1 Score | 89.37 | **91.06** | 90.90 | 88.70 | 80.62 | 80.81 |
> | MRPC | Accuracy | 85.10 | **88.00** | **88.00** | 85.10 | 69.62 | 70.90 |
>
> The VPT using the VTAB-1k [3] benchmark, including 19 datasets categorized into three groups: Natural, Specialized, and Structured. To validate our proposed loss (the PEN loss), we select one dataset from each group. Additionally, we also evaluate the effectiveness of the PEN loss on the MRPC dataset [4], which is a popular benchmark of natural language understanding. The experiment details are based on the GitHub repository at <https://github.com/KMnP/vpt> for VPT with the ViT-B/16 backbone and <https://github.com/huggingface/peft> for prefix-tuning with the RoBERTa-Large backbone.
>
> From presented results, the proposed PEN loss can **enhance the performance** in all conducted experiments, which is an exciting outcome. However, it is acknowledged that determining the ideal weight for the proposed loss requires further investigation, which is a worthy direction that can be explored in future work.
>
> All in all, these preliminary responses our research prospect discussed in the Conclusion Section (i.e. Section 6) and we are hopeful that our finds and approaches will provide new insights to PEFT methods and inspire future research in these fields.
>
> ----
>
> > Q3. Could there be a combination of between soft-prompt tuning and hard-prompt tuning? (hard = explicitly use some predefined words/phrases as part of the prompts)
>
> **A3.** Thanks for your insightful comment considering the possibility of combination of between soft-prompt tuning and hard-prompt tuning. As far as we know, **P-tuning** [5] and **P-tuning v2** [6] have implemented this idea, which employs trainable continuous prompt embeddings in concatenation with discrete prompts, and achieved remarkable performance. We enthusiastically welcome the integration of our method, Nemesis, into P-tuning. We believe that incorporating Nemesis into P-tuning could result in remarkable results.
>
> ----
>
> > Q4. Any idea of further combining existing PEFT (prompt tuning, prefix tuning, LoRA...) with your Nemesis method?
>
> **A4.** The response related to the potential applicable scenarios of Nemesis has provided earlier in **A2**. Our preliminary experiments have demonstrated that our proposed method Nemesis can enhance the performance of visual prompt-tuning and prefix-tuning methods. We sincerely welcome further testing of our approach on more PEFT methods and downstream tasks in the future.
>
>
>
> **References**
>
> [1] Jia, Menglin, et al. Visual prompt tuning. *ECCV*, 2022.
>
> [2] Li, Xiang Lisa, and Percy Liang. Prefix-tuning: Optimizing continuous prompts for generation. *IJCNLP*, 2021.
>
> [3] Zhai, Xiaohua, et al. A large-scale study of representation learning with the visual task adaptation benchmark. *arXiv*, 2019.
>
> [4] William B. Dolan and Chris Brockett. Automatically Constructing a Corpus of Sentential Paraphrases. *IJCNLP*, 2005.
>
> [5] Liu, Xiao, et al. GPT understands, too. *arXiv*, 2021.
>
> [6] Liu, Xiao, et al. P-tuning v2: Prompt tuning can be comparable to fine-tuning universally across scales and tasks. *arXiv*, 2021.

---

### Official Review · Reviewer_hcSQ · 2023-10-30

**Soundness:** 2 fair
**Presentation:** 3 good
**Contribution:** 2 fair
**Rating:** 6
**Confidence:** 3

**Summary:**

The paper, at its core, explores the significant yet uncharted territory around the impact of norms of soft-prompt vectors on the performance of vision-language models (VLMs), like CLIP. The authors have brought to light a unique phenomenon termed the "Low-Norm Effect", highlighting how reducing norms of specific learned prompts can sometimes boost the performance of VLMs. The effect seems to be more prevalent in certain datasets like Imagenet, OxfordPets, and Food101 as compared to others. Interestingly, the Low-Norm Effect appears to have a stronger presence when there's limited training data, hinting at potential issues with soft-prompt methods under data constraints.

To harness this Low-Norm Effect, the paper proposes a method named "Nemesis". This approach introduces two techniques – Position Equality Normalization (PEN) loss and the more refined Position Awareness Normalization (PAN) loss. While the PEN loss aims to normalize the norms of all prompt vectors, the PAN loss is more discerning, identifying positions that might induce the Low-Norm Effect before selectively normalizing them. The authors suggest that this method can notably enhance VLM performance without incurring significant computational costs.

**Strengths:**

1. The paper pioneers a systematic investigation into the role of soft-prompt vector norms in VLMs, addressing a previously unexplored research question.

2. The proposed Nemesis method, with its innovative PEN and PAN losses, offers a potential solution to the Low-Norm Effect, showing promise for improving VLM performance.

3. Extensive corruption experiments shed light on the Low-Norm Effect's impact, providing valuable insights for future soft-prompt tuning endeavors.

**Weaknesses:**

1. $\beta$ can be either 0 or 1, corresponding to two variants of the proposed Nemesis method. However, there is no ablation study on the selection of $\beta$, nor is there an exploration of the potential impact of setting $\beta$ with decimal values to assign weights to the two methods.

2. The paper introduces a pre-inference step before each training batch to identify positions inducing the Low-Norm Effect. Such a step could introduce computational overhead, especially with larger datasets or when rapid training iterations are required. The study hasn’t provided a detailed analysis of the computational cost or time implications this might have in different scenarios.

3. The Position Equality Normalization (PEN) loss applies equal weight to the norms of soft prompts at all positions. While the paper does acknowledge that normalizing prompt vectors at positions unaffected by the Low-Norm Effect may not yield performance improvement, the inherent assumption of the universality of the Low-Norm Effect across positions may not hold true for all datasets or real-world scenarios. The approach could benefit from a more dynamic, adaptive mechanism.

4. The paper utilizes the RESCALE operation with a specific rescaling factor, τ, described as a positive real number less than 1. However, there’s no mention of how the value of τ is determined, if it's consistent across datasets, or its sensitivity. The choice of τ could have implications on the effectiveness of the Nemesis method, and without clear insight into its selection, there’s potential variability in results.

**Questions:**

Given the significance of the parameter $\beta$ in differentiating between the two variants of the Nemesis method, why was an ablation study not conducted to evaluate its impact? Additionally, have you considered exploring decimal values for $\beta$ to potentially strike a balance between the effects of the PEN and PAN losses?

How does the proposed Nemesis method compare with other soft-prompt tuning methods in terms of computational efficiency and scalability, especially in larger datasets or more complex tasks?

---

> ### Author Response · Authors · 2023-11-19
> **Reply for Reviewer hcSQ**
>
> We greatly thank you for your thorough reviews and valuable feedback. We address your concerns as follows.
>
> > Q1. $\beta$ can be either 0 or 1, corresponding to two variants of the proposed Nemesis method. However, there is no ablation study on the selection of $\beta$, nor is there an exploration of the potential impact of setting $\beta$ with decimal values to assign weights to the two methods.
>
> **A1.** We sincerely appreciate your suggestion to conduct ablation study about $\beta$ (i.e. explore the simultaneous combination of the PEN loss and the PAN loss). In response to this concern, we conducted experiments to investigate the combined effect of these two losses on model performance. To be specific, we evaluated various values of $\beta$, including 0, 0.1, 0.3, 0.5, 0.7, 0.9, and 1, where 0 and 1 represent the use of the PAN and PEN loss, respectively. The detailed results are shown in the following table.
> |  | $\beta$=0 | $\beta$=0.1 | $\beta$=0.3 | $\beta$=0.5 | $\beta$=0.7 | $\beta$=0.9 | $\beta$=1 |
> | :---- | :----: | :----: | :----: | :----: | :----: | :----: | :----: |
> | 1-shot | 59.33 | 59.45 | **59.54** | 59.29 | 59.26 | 59.37 | 59.42 |
> | 16-shots | 74.20 | **74.39** | 73.37 | 74.02 | 73.93 | 73.79 | 73.47 |
>
> The results presented in the table are obtained by average results of 11 datasets used in the paper and the best performance is indicated with bold. The results display an evident pattern that combining two proposed normalization losses **improves model performance** compared to using only a single normalization loss, whether it is 1-shot or 16-shots setting. Furthermore, it can be observed that $\beta$=0.3 and $\beta$=0.1 achieve the best results, respectively. This suggests that the PAN loss plays a dominant role, while the PEN loss provides assistance, which can lead to improved performance.
>
> We apologize for not discussing the ablation studies of $\beta$ in the paper. We understand the importance of including this information, so we have added it into the Ablation Study Section (i.e. Section 4.7) of the updated paper.
>
> ----
>
> > Q2. The paper introduces a pre-inference step before each training batch to identify positions inducing the Low-Norm Effect. Such a step could introduce computational overhead, especially with larger datasets or when rapid training iterations are required. The study hasn’t provided a detailed analysis of the computational cost or time implications this might have in different scenarios.
>
> **A2.** We recognize your concern regarding a detailed analysis of the computation costs raised by our method Nemesis. Actually, we provided the analysis of computation costs on the FGVCAircraft dataset in the Ablation Studies Section of the original version. For more comprehensive understanding of the computation costs raised by Nemesis, we conducted experiments across 11 datasets used in the paper. Additionally, we also compared our method with **PLOT** [1], which is a state-of-the-art method for prompt-tuning VLMs, from the aspect of computation costs. The detailed results can be found in the following tables.
>
> | Datasets     | Classes | CoOp             | CoOp+PEN            | CoOp+PAN            | CoOp+PEN+PAN        | PLOT                |
> | ------------ | ------- | ---------------- | ------------------- | ------------------- | ------------------- | ------------------- |
> | ImageNet     | 1000    | 1$\times$ (0.22) | 2.27$\times$ (0.50) | 4.00$\times$ (0.88) | 4.05$\times$ (0.89) | -                   |
> | SUN397       | 397     | 1$\times$ (0.11) | 1.91$\times$ (0.21) | 3.36$\times$ (0.37) | 3.45$\times$ (0.38) | 3.00$\times$ (0.33) |
> | StanfordCars | 196     | 1$\times$ (0.07) | 1.57$\times$ (0.11) | 2.86$\times$ (0.20) | 2.86$\times$ (0.20) | 2.57$\times$ (0.18) |
> | Flowers102   | 102     | 1$\times$ (0.06) | 1.17$\times$ (0.07) | 2.00$\times$ (0.12) | 2.00$\times$ (0.12) | 1.83$\times$ (0.11) |
> | UCF101       | 101     | 1$\times$ (0.06) | 1.17$\times$ (0.07) | 2.00$\times$ (0.12) | 2.17$\times$ (0.13) | 1.83$\times$ (0.11) |
> | Food101      | 101     | 1$\times$ (0.06) | 1.17$\times$ (0.07) | 2.00$\times$ (0.12) | 2.00$\times$ (0.12) | 1.83$\times$ (0.11) |
> | Caltech101   | 100     | 1$\times$ (0.06) | 1.17$\times$ (0.07) | 2.00$\times$ (0.12) | 2.00$\times$ (0.12) | 1.83$\times$ (0.11) |
> | FGVCAircraft | 100     | 1$\times$ (0.06) | 1.17$\times$ (0.07) | 2.00$\times$ (0.12) | 2.00$\times$ (0.12) | 1.83$\times$ (0.11) |
> | DTD          | 47      | 1$\times$ (0.05) | 1.00$\times$ (0.05) | 1.60$\times$ (0.08) | 1.60$\times$ (0.08) | 1.40$\times$ (0.07) |
> | OxfordPets   | 37      | 1$\times$ (0.05) | 1.00$\times$ (0.05) | 1.40$\times$ (0.07) | 1.60$\times$ (0.08) | 1.40$\times$ (0.07) |
> | EuroSAT      | 10      | 1$\times$ (0.05) | 1.00$\times$ (0.05) | 1.20$\times$ (0.06) | 1.20$\times$ (0.06) | 1.00$\times$ (0.05) |

---

> ### Author Response · Authors · 2023-11-19
> **Reply for Reviewer hcSQ**
>
> The first table provides results about computation costs of the benchmark CoOp, our two losses, and PLOT. For fair comparison, we do not report the computation costs of PLOT on the ImageNet dataset as it needs to reduce batch size for single GPU execution. The PAN loss adopts the default setting $N$ = 1. All comparison results are obtained by the average of multiple training batches. The numbers in parentheses indicate the corresponding running times in seconds. **Firstly**, we can observe that incorporating the PEN or PAN loss increases computation costs compared to using CoOp alone. The PAN loss introduces an additional pre-inference step before each training batch, resulting in a higher computational burden compared to the PEN loss. Combining both losses does not significantly increase the running time compared to using the PAN loss alone. **Furthermore**, there is a clear and consistent trend across all methods, including CoOp, PLOT, and our two losses, where computation costs increase with the number of classes in the datasets. We think that this is caused by the fact that CoOp-based methods optimize the similarity scores between text features of all categories and image features of a mini-batch. The increasing of number of classes will result in a significant increase in the gradients required for calculation. **Lastly**, it should be noted that the running time of the PAN loss and PLOT is almost comparable.
>
> | Datasets     | CoOp             | CoOp+PAN($N$=1)     | CoOp+PAN($N$=2)     | CoOp+PAN($N$=4)     | CoOp+PAN($N$=8)      | CoOp+PAN($N$=16)     |
> | ------------ | ---------------- | ------------------- | ------------------- | ------------------- | -------------------- | -------------------- |
> | ImageNet     | 1$\times$ (0.22) | 4.00$\times$ (0.88) | 4.86$\times$ (1.07) | 6.64$\times$ (1.46) | 10.09$\times$ (2.22) | 17.05$\times$ (3.75) |
> | SUN397       | 1$\times$ (0.11) | 3.36$\times$ (0.37) | 4.09$\times$ (0.45) | 5.36$\times$ (0.59) | 8.18$\times$ (0.90)  | 13.64$\times$ (1.50) |
> | StanfordCars | 1$\times$ (0.07) | 2.86$\times$ (0.20) | 3.43$\times$ (0.24) | 4.43$\times$ (0.31) | 6.57$\times$ (0.46)  | 10.71$\times$ (0.75) |
> | Flowers102   | 1$\times$ (0.06) | 2.00$\times$ (0.12) | 2.50$\times$ (0.15) | 3.00$\times$ (0.18) | 4.43$\times$ (0.26)  | 6.83$\times$ (0.41)  |
> | UCF101       | 1$\times$ (0.06) | 2.00$\times$ (0.12) | 2.33$\times$ (0.14) | 3.00$\times$ (0.18) | 4.43$\times$ (0.26)  | 6.83$\times$ (0.41)  |
> | Food101      | 1$\times$ (0.06) | 2.00$\times$ (0.12) | 2.33$\times$ (0.14) | 3.00$\times$ (0.18) | 4.43$\times$ (0.26)  | 6.67$\times$ (0.40)  |
> | Caltech101   | 1$\times$ (0.06) | 2.00$\times$ (0.12) | 2.33$\times$ (0.14) | 3.00$\times$ (0.18) | 4.17$\times$ (0.25)  | 6.67$\times$ (0.40)  |
> | FGVCAircraft | 1$\times$ (0.06) | 2.00$\times$ (0.12) | 2.33$\times$ (0.14) | 3.00$\times$ (0.18) | 4.17$\times$ (0.25)  | 6.67$\times$ (0.40)  |
> | DTD          | 1$\times$ (0.05) | 1.60$\times$ (0.08) | 1.80$\times$ (0.09) | 2.20$\times$ (0.11) | 2.80$\times$ (0.14)  | 4.20$\times$ (0.21)  |
> | OxfordPets   | 1$\times$ (0.05) | 1.40$\times$ (0.07) | 1.80$\times$ (0.09) | 1.80$\times$ (0.09) | 2.40$\times$ (0.12)  | 3.40$\times$ (0.17)  |
> | EuroSAT      | 1$\times$ (0.05) | 1.20$\times$ (0.06) | 1.20$\times$ (0.06) | 1.20$\times$ (0.06) | 1.40$\times$ (0.07)  | 1.80$\times$ (0.09)  |
>
> The results in the second table highlight that increasing the value of $N$ leads to increased computational costs, particularly dealing with large datasets. This is because the model requires generating a greater number of corrupted textual features for prediction within each training batch. Although it is generally observed that increasing $N$ has a positive impact on the model's performance, finding ways to mitigate the computational burden is an interesting direction for future research. In practical applications, we suggest using $N$ = 1 to strike a balance between computation costs and performance improvement.
>
> To sum up, despite increased computational costs associated with our method, **the running speed is still fast even for large-scale datasets**. We **have added a new subsection** to discuss computation costs, which can be found in Appendix A.2.7 of the updated paper.

---

> ### Author Response · Authors · 2023-11-19
> **Reply for Reviewer hcSQ**
>
> > Q3. The Position Equality Normalization (PEN) loss applies equal weight to the norms of soft prompts at all positions. While the paper does acknowledge that normalizing prompt vectors at positions unaffected by the Low-Norm Effect may not yield performance improvement, the inherent assumption of the universality of the Low-Norm Effect across positions may not hold true for all datasets or real-world scenarios. The approach could benefit from a more dynamic, adaptive mechanism.
>
> **A3.** We appreciate your insightful comment on our proposed PEN loss. As your statement, there is an inherent assumption of the diversity of the Low-Norm Effect across positions, which aligns with our observations during experiments. The mechanism behind this diversity is complex as the positions that induce the Low-Norm Effect can change over time during training. Hence, the PEN loss seems be not sufficiently general and adaptive because it assigns equal weight to the norms of soft prompts at all positions. Nevertheless, this does not impede the PEN loss from being employed in more dynamic and adaptive forms. For instance, it is viable to randomly normalize soft prompts at one or more positions or apply random sizes of normalization for various positions. According to our ablation studies in the paper (i.e. Table 3), it is evident that randomly normalizing partial positions outperforms normalizing all positions. Therefore, we acknowledge your comment suggesting that the PEN loss could benefit from a more dynamic adaptive mechanism.
>
> On the other hand, it actually motivates us to propose **the PAN loss as a more dynamic and adaptive alternative** to the PEN loss. The PAN loss can effectively identify the prompt positions that induce the Low-Norm Effect and then normalize them accordingly. Furthermore, our empirical results illustrate that the PAN loss does perform better in a large shot setting.
>
> Of course, it is encouraged to investigate other adaptive alternatives in our future work.
>
> ----
>
> > Q4. The paper utilizes the RESCALE operation with a specific rescaling factor, $\tau$, described as a positive real number less than 1. However, there’s no mention of how the value of $\tau$ is determined, if it's consistent across datasets, or its sensitivity. The choice of $\tau$ could have implications on the effectiveness of the Nemesis method, and without clear insight into its selection, there’s potential variability in results.
>
> **A4.** Thank you for your comment regarding the discussion of the rescaling factor $\tau$ used in the PAN loss. We would like to kindly point out that it seems to have been overlooked in your review. In fact, the paper did provide hyper-parameter analysis for different $\tau$ values, including 0.1, 0.5, and 0.9. Based on our results, $\tau$ = 0.9 achieved slightly superior performance on average across multiple datasets compared to the default value of 0.5 that was initially used. Generally, the value of $\tau$ **ranging from 0.1 to 0.9 did not significantly impact** the model's performance.
>
> However, we have not tested smaller values of $\tau$, such 0.01 and 0.001, when using the PAN loss. Based on the results of our corruption experiments (i.e. Table A3), it is observed that smaller values of $\tau$ typically lead to a lower occurrence frequency of the Low-Norm Effect. This suggests that extremely small values of $\tau$ might affect the effectiveness of the PAN loss because they cannot identify more prompt positions that could induce the Low-Norm Effect during the training process. Therefore, we suggest **using 0.5 or 0.9** as the value of $\tau$ when employing the Nemesis method.
>
> ----
>
> > Q5. Given the significance of the parameter $\beta$ in differentiating between the two variants of the Nemesis method, why was an ablation study not conducted to evaluate its impact? Additionally, have you considered exploring decimal values for $\beta$ to potentially strike a balance between the effects of the PEN and PAN losses?
>
> **A5.** The response related to $\beta$ has provided earlier in **A1**. As suggested, we have incorporated an ablation study about $\beta$ into the Ablation Study Section (i.e. Section 4.7) of the updated paper.

---

> ### Author Response · Authors · 2023-11-19
> **Reply for Reviewer hcSQ**
>
> > Q6. How does the proposed Nemesis method compare with other soft-prompt tuning methods in terms of computational efficiency and scalability, especially in larger datasets or more complex tasks?
>
> **A6.** The response related to computation costs has provided earlier in **A2**. As suggested, we have added a subsection (i.e. Appendix A.2.7) to analyze computation costs in the revised paper.
>
> As for **scalability**, we have conducted preliminary experiments on a few PEFT methods and their applicable scenarios, including **visual prompt-tuning (VPT)** [2] for image classification and **prefix-tuning** [3] for paraphrase classification. Due to time constraints during the rebuttal period, we focused exclusively on testing our proposed PEN loss, which can be seamlessly and effortlessly integrated into existing PEFT methods. From presented results in **A3 of Reviewer KnC5** or **A2 of Reviewer bzG1**, the proposed PEN loss **can enhance the performance** in all conducted experiments, which is an exciting outcome. On the other hand, the proposed PAN loss may involve designing specific corruption experiments and adopting distinct performance comparison metrics depending on different methods and tasks. To be more specific, we can adopt various performance metrics when identifying the positions that induce the Low-Norm Effect, such as classification accuracy for classification tasks, mAP (mean Average Precision) for object detection tasks, IoU (Intersection over Union) for segmentation tasks, and BLEU (BilinguaL Evaluation Understudy) for text generation. We hope to conduct more research to explore its applicability and adaptability in various PEFT tasks in our future work.
>
>
>
> **References**
>
> [1] Chen, Guangyi, et al. PLOT: Prompt Learning with Optimal Transport for Vision-Language Models. *ICLR*, 2023.
>
> [2] Jia, Menglin, et al. Visual prompt tuning. *ECCV*, 2022.
>
> [3] Li, Xiang Lisa, and Percy Liang. Prefix-tuning: Optimizing continuous prompts for generation. *IJCNLP*, 2021.

---

> > ### Comment · Reviewer_hcSQ · 2023-11-19
> > **Re Authors**
> >
> > Thank you to the authors for their diligent rebuttal and the additional experiments, which may could support the claims in this paper and address my main concerns. I have improved my score for this work.

---

> > > ### Author Response · Authors · 2023-11-23
> > > **Reply for Reviewer hcSQ**
> > >
> > > Dear Reviewer hcSQ,
> > >
> > > Thank you for taking the time and effort to review our research work. We are pleased to address your concerns. Additionally, we sincerely appreciate the increase in score and your recognition of the value of our paper.
> > >
> > > Wish you all the best!
> > >
> > > Best regards,
> > >
> > > Authors

---

### Official Review · Reviewer_KnC5 · 2023-11-05

**Soundness:** 4 excellent
**Presentation:** 3 good
**Contribution:** 3 good
**Rating:** 8
**Confidence:** 4

**Summary:**

The paper discussed the influence of soft-prompt to VLM, introduced REPLACE and RESCALE corruption affecting VLM, and proposed two normalization loss improving the performance of soft-prompt. The authors conducted a lot of experiments to confirm the effectiveness of method.

**Strengths:**

1、The paper is the first study to discuss the influence of soft-prompt toward VLM.
2、The paper conducted REPLACE and RESCALE to discuss the normalization of soft-prompt, and proposed Nemesis including two normalization losses to improve the effectiveness of soft-prompt.
3、The paper has conducted a lot of experiments to prove the effectiveness of the method.

**Weaknesses:**

1、The writing of some parts of the paper are not clear enough. It is recommended that the authors check. For example, there is a discrepancy between formula 4 and the symbol definition in the previous paragraph.
2、The two types of losses proposed in the paper lack a correlation with practical significance, suggesting authors discuss why the two forms of normalization affect soft prompt.
3、The paper lacks discussion on the applicable scenarios of two normalization losses.

**Questions:**

1、The paper proposes two normalization methods, while only testing the effects of PEN and PAN on the experimental results respectively. Why cannot both types of losses be used simultaneously? If there is a contradiction between the two losses, it is recommended that the authors discuss the differences. If the two losses are similar, can the two losses be unified? If the two losses gain from different perspective, should relevant experiments be provided?
2、Can author discuss application circumstance of two normalization methods? In practical applications, what kind of normalization loss should we choose for what situation? Suggest the authors to discuss.

---

> ### Author Response · Authors · 2023-11-19
> **Reply for Reviewer KnC5**
>
> Thanks for your positive comments and valuable feedback. We address your concerns as follows.
>
> ----
>
> > Q1. The writing of some parts of the paper are not clear enough. It is recommended that the authors check. For example, there is a discrepancy between formula 4 and the symbol definition in the previous paragraph.
>
> **A1.** Thank you for your attention and feedback. We have carefully reviewed the symbol definitions and have made the necessary revisions to address the issue you mentioned. In order to differentiate the subscripts of $\alpha$ between in Eq. (3) and in Eq. (4), we have revised Eq. (4) in the updated paper.
>
> ----
>
> > Q2. The two types of losses proposed in the paper lack a correlation with practical significance, suggesting authors discuss why the two forms of normalization affect soft prompt.
>
> **A2.** We apologize for not explicitly discussing the specific impacts of two normalization losses on models in the paper. However, we have added a subsection in the Appendix (i.e. Appendix A.2.6) of the revised paper to discuss this. We hope that it can address your concern.
>
> Taking the Caltech101 dataset [1] as an example, we conducted experiments to illustrate the impact of two proposed losses on soft prompts and model performance. Here, we straightforwardly shows several important observations obtained by experimental results. **Firstly**, regardless of whether it is in the 1-shot or 16-shot setting, the PEN loss demonstrates a stronger level of normalization compared to the PAN loss. On the other hand, despite the stronger normalization of the PEN loss, it does not outperform the PAN loss in the 16-shot setting. This implies that simply reducing the norm of soft prompts does not always lead to an improvement in model performance. **Secondly**, there is a notable occurrence frequency of the Low-Norm Effect in the 16-shot setting during the training process, indicating that an increasing number of training samples can facilitate the identification of prompting positions that induce the Low-Norm Effect by the PAN loss. This observation may explain why the PAN loss performs better in the 16-shot setting but is inferior to the PEN loss in the 1-shot setting. **Lastly**, the addition of two normalization losses does not seem to have a substantial influence on the original cross-entropy loss, suggesting that they do not contribute to the model performance primarily by reducing the cross-entropy loss. Instead, their benefits are likely derived from harnessing the Low-Norm Effect. The visual results can be found in the updated paper.

---

> ### Author Response · Authors · 2023-11-19
> **Reply for Reviewer KnC5**
>
> > Q3. The paper lacks discussion on the applicable scenarios of two normalization losses.
>
> **A3.** We appreciate your concern regarding the need for discussion on potential applicable scenarios. While our proposed method primarily focuses on benchmarking soft prompt-tuning VLMs and several downstream VLM-based tasks, including few-shot image classification, domain generalization, and base-to-new generalization, it should be noted that the benefits of the two proposed normalization losses may be not limited to these tasks alone. We speculate that they can also be applied to other parameter-efficient tuning (**PEFT**) methods, such as prompt-tuning, prefix-tuning, P-tuning, adapter-tuning, and LoRA, as well as their various downstream tasks, including vision-related tasks like image classification, object detection and semantic segmentation, as well as language-related tasks like text classification and generation.
>
> To verify this, we conducted preliminary experiments on a few PEFT methods and their applicable scenarios, including **visual prompt-tuning (VPT)** [2] for image classification and **prefix-tuning** [3] for paraphrase classification. Due to time constraints during the rebuttal period, we focused exclusively on testing our proposed PEN loss, which can be seamlessly and effortlessly integrated into existing PEFT methods. On the other hand, the proposed PAN loss may involve designing specific corruption experiments and adopting distinct performance comparison metrics depending on different methods and tasks. To be more specific, we should adopt various performance metrics when identifying the positions that induce the Low-Norm Effect, such as classification accuracy for classification tasks, mAP (mean Average Precision) for object detection tasks, IoU (Intersection over Union) for segmentation tasks, and BLEU (BilinguaL Evaluation Understudy) for text generation. We hope for more research to explore its applicability and adaptability in various PEFT tasks in the future. Hence, here we only report our preliminary results based on the PEN loss using various values of $\omega$, as shown in following tables.  $\omega$ is a scaling coefficient that controls the normalization strength.
> | Datasets | $\omega$=0 | $\omega$=0.001 | $\omega$=0.01 | $\omega$=0.1 | $\omega$=1 | $\omega$=10 |
> | :---- | :----: | :----: | :----: | :----: | :----: | :----: |
> | Sun397 (Natural) | 49.86 | 49.76 | **50.04** | 48.89 | 46.67 | 45.07 |
> | EuroSAT (Specialized) | 91.89 | 91.30 | 92.00 | **92.81** | 91.63 | 91.31 |
> | DMLab (Structured) | 33.54 | 33.39 | 33.75 | 34.85 | **35.75** | 35.50 |
>
> | Datasets | Performance Metrics | $\omega$=0 | $\omega$=0.0001 | $\omega$=0.001 | $\omega$=0.01 | $\omega$=0.1 | $\omega$=1 |
> | :---- | :----: | :----: | :----: | :----: | :----: | :----: | :----: |
> | MRPC | F1 Score | 89.37 | **91.06** | 90.90 | 88.70 | 80.62 | 80.81 |
> | MRPC | Accuracy | 85.10 | **88.00** | **88.00** | 85.10 | 69.62 | 70.90 |
>
> The VPT using the VTAB-1k [4] benchmark, including 19 datasets categorized into three groups: Natural, Specialized, and Structured. To validate our proposed loss (the PEN loss), we select one dataset from each group. Additionally, we also evaluate the effectiveness of the PEN loss on the MRPC dataset [5], which is a popular benchmark of natural language understanding. The experiment details are based on the GitHub repository at <https://github.com/KMnP/vpt> for VPT with the ViT-B/16 backbone and <https://github.com/huggingface/peft> for prefix-tuning with the RoBERTa-Large backbone.
>
> From presented results, the proposed PEN loss can **enhance the performance** in all conducted experiments. However, it is acknowledged that determining the ideal weight for the proposed loss requires further investigation, which could be a worthy direction for our future work.
>
> In summary, those preliminary responses our research prospect discussed in the Conclusion Section (i.e. Section 6) and we hope that our finds and approaches will provide new insights to PEFT methods and inspire future research in these fields.

---

> ### Author Response · Authors · 2023-11-19
> **Reply for Reviewer KnC5**
>
> > Q4. The paper proposes two normalization methods, while only testing the effects of PEN and PAN on the experimental results respectively.
>
> **A4.** We sincerely appreciate your suggestion to explore the simultaneous combination of the PEN loss and the PAN loss. In response to this concern, we conducted experiments to investigate the combined effect of these two losses on model performance. To be specific, we evaluated various values of $\beta$, including 0, 0.1, 0.3, 0.5, 0.7, 0.9, and 1, where 0 and 1 represent the use of the PAN and PEN loss, respectively. The detailed results are shown in the following table.
> |  | $\beta$=0 | $\beta$=0.1 | $\beta$=0.3 | $\beta$=0.5 | $\beta$=0.7 | $\beta$=0.9 | $\beta$=1 |
> | :---- | :----: | :----: | :----: | :----: | :----: | :----: | :----: |
> | 1-shot | 59.33 | 59.45 | **59.54** | 59.29 | 59.26 | 59.37 | 59.42 |
> | 16-shots | 74.20 | **74.39** | 73.37 | 74.02 | 73.93 | 73.79 | 73.47 |
>
> The results presented in the table are obtained by average results of 11 datasets used in the paper and the best performance is shown in bold. The results display an evident pattern that combining two proposed normalization losses **improves model performance** compared to using only a single normalization loss, whether it is 1-shot or 16-shots setting. Furthermore, it can be observed that $\beta$=0.3 and $\beta$=0.1 achieve the best results, respectively. This suggests that the PAN loss plays a dominant role, while the PEN loss provides assistance, which can lead to improved performance.
>
> We apologize for not discussing the ablation studies of $\beta$ in the paper. We understand the importance of including this information, so we have added it into the Ablation Study Section (i.e. Section 4.7) of the updated paper.
>
> ----
>
> > Q5. Can author discuss application circumstance of two normalization methods? In practical applications, what kind of normalization loss should we choose for what situation? Suggest the authors to discuss.
>
> **A5.** Thank you for highlighting this concern. This paper proposed two types of normalization losses to harness the Low-Norm Effect during soft prompt-tuning vision-language models. Based on the few-shot recognition results presented in the first subfigure of Figure 2 and Table A5, it can be observed that the PEN loss generally outperforms the PAN loss for a small number of training examples (i.e., 1, 2, and 4 shots). On the other hand, the PAN loss exhibits superior performance compared to the PEN loss when using a larger number of training samples (i.e., 8 and 16 shots). This suggests that the PEN loss can be employed for settings with small shots while the PAN loss is more suitable for large shots.
>
> Additionally, based on the domain generalization results presented in Table 1, we can observe that the PEN loss performs better than the PAN loss in target domains, while the latter shows superior performance in source domains compared to the former. This implies that the PEN loss is more robust for domain shifts.
>
> To sum up, **our recommendation** is to utilize the PEN loss for training samples with small shots or when handling domain shifts. Meanwhile, employing the PAN loss can significantly enhance the model's performance on intra-domain data with large shots.
>
>
>
> **References**
>
> [1] Fei-Fei, Li, et al. One-shot learning of object categories. *TPAMI*, 2006.
>
> [2] Jia, Menglin, et al. Visual prompt tuning. *ECCV*, 2022.
>
> [3] Li, Xiang Lisa, and Percy Liang. Prefix-tuning: Optimizing continuous prompts for generation. *IJCNLP*, 2021.
>
> [4] Zhai, Xiaohua, et al. A large-scale study of representation learning with the visual task adaptation benchmark. *arXiv*, 2019.
>
> [5] William B. Dolan and Chris Brockett. Automatically Constructing a Corpus of Sentential Paraphrases. *IJCNLP*, 2005.

---

### Author Response · Authors · 2023-11-19
**General Reply**

Dear Reviewers,

We sincerely appreciate your constructive comments and valuable feedback.

We have responded to each reviewer separately, answering your questions one-by-one. We believe that our responses adequately address your concerns.

We address two major concerns as follows.

----

> Q1. Any more applicable scenarios about the proposed method? Any idea of further combining existing PEFT (prompt tuning, prefix tuning, LoRA...) with your Nemesis method?

**A1.** We appreciate Reviewers' concerns regarding the need for discussion on potentially applicable scenarios, especially further combining existing PEFT with our proposed method Nemesis. We believe that it can be applied to other parameter-efficient tuning (**PEFT**) methods, such as prompt-tuning, prefix-tuning, P-tuning, adapter-tuning, and LoRA, as well as their various downstream tasks, including vision-related tasks like image classification, object detection and semantic segmentation, as well as language-related tasks like text classification and generation.

To verify this, we conducted preliminary experiments on a few PEFT methods and their applicable scenarios, including **visual prompt-tuning (VPT)** [1] for image classification and **prefix-tuning** [2] for paraphrase classification. The results demonstrate the proposed PEN loss can **enhance the performance** in all conducted experiments, which is an exciting outcome. Due to time constraints during the rebuttal period, we have not conducted experiments using the PAN loss and on more PEFT methods, but we sincerely welcome further testing of our approach on more PEFT methods and downstream tasks in the future. More details about these two experiments can be found in **A3 of Reviewer KnC5** or **A2 of Reviewer bzG1**.

----

> Q2. Why not use both types of losses (i.e. PEN and PAN) simultaneously?

**A2.** We apologize for not discussing the ablation studies of $\beta$ in the paper. We understand the importance of including this information, so we **have added it** into the Ablation Study Section (i.e. Section 4.7) of the revision. The results were obtained by 11 datasets, as shown in the following Table.

|          | $\beta$=0 | $\beta$=0.1 | $\beta$=0.3 | $\beta$=0.5 | $\beta$=0.7 | $\beta$=0.9 | $\beta$=1 |
| :------- | :-------: | :---------: | :---------: | :---------: | :---------: | :---------: | :-------: |
| 1-shot   |   59.33   |    59.45    |  **59.54**  |    59.29    |    59.26    |    59.37    |   59.42   |
| 16-shots |   74.20   |  **74.39**  |    73.37    |    74.02    |    73.93    |    73.79    |   73.47   |

The best performance is indicated with bold. It is not hard to find that combining two proposed normalization losses **improves model performance** compared to using only a single normalization loss. Furthermore, it can be observed that $\beta$=0.3 and $\beta$=0.1 achieve the best results, respectively. This suggests that the PAN loss plays a dominant role, while the PEN loss provides assistance, which can lead to improved performance. We sincerely appreciate reviewers' advice to discuss the possibility of combining both losses, which makes our work more complete and better.

----

We have incorporated the suggestions provided by the reviewers and made revisions accordingly. In the updated paper, we have included additional discussions and highlighted the altered or newly added text, figures, and tables in **blue**. This highlighting will facilitate easier observation and identification of the changes made. If you have any further concern or question, please let us know. Thank you once again for your time and effort.

Best regards,

Authors



**References**

[1] Jia, Menglin, et al. Visual prompt tuning. *ECCV*, 2022.

[2] Li, Xiang Lisa, and Percy Liang. Prefix-tuning: Optimizing continuous prompts for generation. *IJCNLP*, 2021.

---

### Meta-Review · Area_Chair_2LCH · 2023-12-17

**Metareview:**

This paper identifies the Low-Norm Effect phenomenon in soft prompts of vision-language models (VLMs). The authors propose Nemesis, a pioneering solution using novel normalization techniques (PEN and PAN). The paper presents comprehensive experiments to validate its effectiveness and provides insights for future soft-prompt tuning. The reviewers were largely positive about this work. However, there were concerns about clarity and a lack of discussions on the practical significance of the work, which I believe the authors have addressed in the new version. There are also major concerns, such as the universality assumption of the Low-Norm Effect across positions, which may not hold for all datasets or real-world scenarios. The authors have offered new insights in the discussion, but it is recommended that they also include such discussions in the revised paper.

**Justification For Why Not Higher Score:**

The paper does not seem sufficiently strong to justify an oral presentation.

**Justification For Why Not Lower Score:**

I believe this paper should be of interest to a wide audience who work on both vision and language.

---

### Decision · Program_Chairs · 2024-01-16

Accept (spotlight)